# Optogenetic spatial patterning of cooperation in yeast populations

Matthias Le Bec [1], Sylvain Pouzet [1], Céline Cordier [1], Simon Barral[1], Vittore Scolari [1,2], Benoit Sorre [1], Alvaro Banderas [1,3] ✉ & Pascal Hersen [1,3] ✉

Microbial communities are shaped by complex metabolic interactions such as cooperation and competition for resources. Methods to control such interactions could lead to major advances in our ability to better engineer microbial consortia for synthetic biology applications. Here, we use optogenetics to control SUC2 invertase production in yeast, thereby shaping spatial assortment of cooperator and cheater cells. Yeast cells behave as cooperators (*i.e.*, transform sucrose into hexose, a public good) upon blue light illumination or cheaters (*i.e.*, consume hexose produced by cooperators to grow) in the dark. We show that cooperators benefit best from the hexoses they produce when their domain size is constrained between two cut-off length-scales. From an engineering point of view, the system behaves as a bandpass filter. The lower limit is the trace of cheaters' competition for hexoses, while the upper limit is defined by cooperators' competition for sucrose. Cooperation mostly occurs at the frontiers with cheater cells, which not only compete for hexoses but also cooperate passively by letting sucrose reach cooperators. We anticipate that this optogenetic method could be applied to shape metabolic interactions in a variety of microbial ecosystems.

Metabolic interactions, such as competition and cooperation to metabolize nutrients, are central to the development of microbial colonies and biofilms[1–4]. These metabolic interactions participate in the establishment of complex, spatially structured multicellular systems in which cells located at different positions experience varied microenvironments and can compete or cooperate with each other. Cooperating cells (cooperators) are defined by their capacity to invest resources to promote the proliferation (i.e., increase the fitness) of other cells[5–7]. Conversely, cells that benefit from other cells without contributing to their metabolic efforts are defined as cheaters[8–11] since they compete for resources without paying the cost required to produce the resources.

Cooperation and competition dynamics have been studied in various contexts[12,13], particularly using the canonical example of sucrose utilization by the budding yeast *Saccharomyces cerevisiae*[14–18]

(Fig. 1). Briefly, yeast cells can produce the invertase Suc2p, which is retained in the periplasmic space and can catalyze the hydrolysis of sucrose into glucose and fructose. Both of these hexoses (glucose and fructose) can then be taken up by cells and metabolized intracellularly (Fig. 1a, Supplementary Fig. 1). Since hydrolysis of sucrose occurs in the periplasm, the hexoses produced by hydrolysis are not only taken up but also leaked into the extracellular space and become public goods as these sugars can also diffuse away and be consumed by adjacent cells (including cheater cells). This situation both favors interactions between individual cells and also represents a fitness cost for the cells expressing Suc2p[16]. Note that although sucrose can alternatively be taken up via sucrose-proton symporters (Mal11p, Mal31p, and Mph2/3p)[19] to then be hydrolyzed internally (a cytosolic form of the invertase, maltase, and isomaltase), usually external hydrolysis is the dominant sucrose uptake process in wild-type yeast[20,21].

[1]Institut Curie, Université PSL, Sorbonne Université, CNRS UMR168, Laboratoire Physico Chimie Curie, 75005 Paris, France. [2]Institut Curie, Université PSL, Sorbonne Université, CNRS UMR3664, Laboratoire Dynamique du Noyau, 75005 Paris, France. [3]These authors contributed equally: Alvaro Banderas, Pascal Hersen. ✉e-mail: alvaro.banderas@curie.fr; pascal.hersen@curie.fr

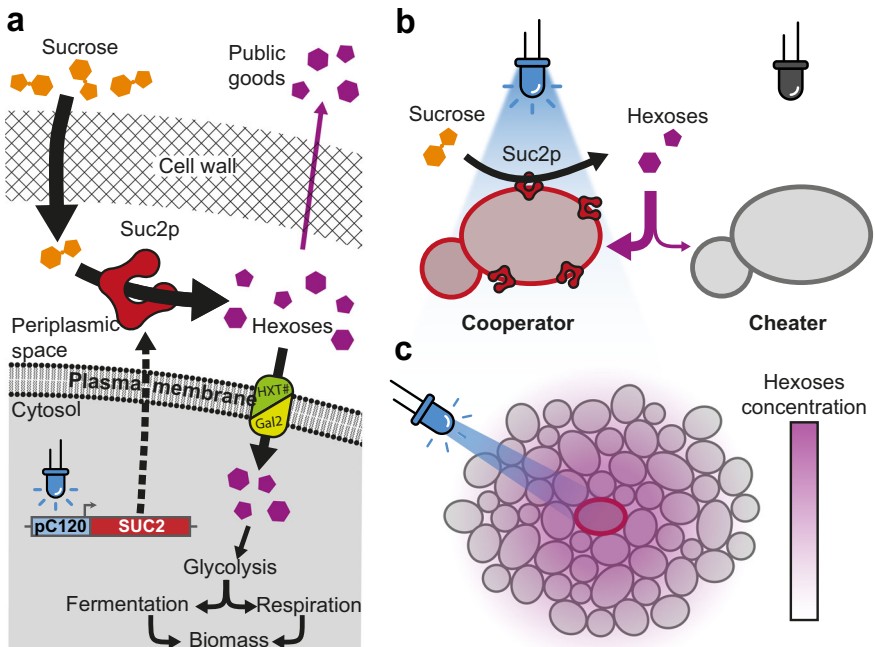

**Fig. 1 | Optogenetic control of yeast sucrose catabolism. a** Blue light illumination induces transcription of the *SUC2* gene and the production of the invertase Suc2p, which is secreted by exocytosis and retained in the periplasm. There, Suc2p catalyzes the hydrolysis of sucrose into two hexoses (glucose and fructose). These hexoses can be imported by cells via specific transporters (HTX1–4, 6–7, and Gal2) to support the growth of yeast cells through glycolysis. Alternatively, glucose and fructose can also diffuse away from the producing cell into the extracellular environment. **b** If the optogenetic system is tightly controlled, only cells stimulated by light can produce Suc2p, while cells in the dark cannot produce Suc2p. Projecting patterns of light on a yeast assembly induces well-separated spatial domains of cooperators and cheaters: illuminated cells behave as cooperators (i.e., they produce hexoses as public goods), while cells in the dark behave as cheaters (i.e., they rely on cooperators' production of public goods to grow). **c** Illumination induces the local production of hexoses and the establishment of hexose gradients through diffusion and uptake by both cooperators and cheaters.

While it is now well accepted in the literature that spatial structure plays a determinant role in natural communities' fate[4,22–25]; most controlled laboratory experiments pursuing a quantitative understanding of microbial competition/cooperation mechanisms have focused on well-mixed or small-scale populations. For example, the seminal works by Maclean et al.[26] compared the stability of cooperation in yeast cooperator-cheater cocultures and further used a 96-well plate to mimic the spatial structure of a *metapopulation*. Although this experimental system was key to studying the impact of the cheater/cooperator ratio on the global population fitness, it cannot be used to create spatially extended and interacting domains of cheaters and cooperators. In fact, it is hard to experimentally create, in a controlled manner, a microbial community of cheaters and cooperators with a user-defined spatial assortment[24] that would allow us to explore quantitatively its impact on microbial cooperation. Here, we propose to use optogenetics to solve this issue and further experimentally explore the microbial interactions over scales of spatial assortment.

In the case of sucrose utilization by structured yeast populations, the key parameters that define the length scales at which cooperation and competition occur are the diffusion and consumption of both hexoses and sucrose. Simple estimates based on the diffusion–reaction equation indicate that the typical length scale over which the concentration decreases from a point source at steady state varies as the square root of the diffusion coefficient of the metabolite of interest and the inverse of the square root of the rate at which the metabolite is absorbed by cells. In expanding microbial colonies, this distance over which metabolic gradients are formed is of the order of a few hundred micrometers[27]. The absorption rate ($k$) depends on the local cell density, which also varies in time as hexoses are progressively transformed into biomass. Hence, the cell density of cooperators and cheaters is likely to be a key parameter that defines the cooperation potency and indeed, this parameter has been shown to be critical in the case of sucrose cooperation dynamics. For example, in liquid culture, higher population growth was observed when yeast formed multicellular clumps[16]. Other researchers showed that—in silico—static growth assortment (where daughter cells stay close to their mother) stabilizes yeast cooperation for sucrose catabolism[28]. Taken together, these results suggest that cooperation through the production of a diffusible public good appears generally more cost-efficient if the cooperating cells stay close together[3].

However, most experimental studies have not considered the impact of the spatial organization of cheating and cooperating domains on the dynamics of cooperation, and, so far, very few experimental quantitative studies[29,30] have investigated this problem, mostly due to technological limitations. Optogenetics, because it permits the achievement of quantitative, spatial control of gene expression over a population of microorganisms, sounds like a tool of choice to address those questions. To date, optogenetics has been reported in only a few works as a tool to spatially control public goods, such as the production of extracellular matrix in *Sinorhizobium meliloti*[31], of adhesin in *Escherichia coli*[32] bacteria or of the SUC2 invertase in budding yeast[29] in recent work by Moreno Morales et al.[29]. These studies used the potential of spatial patterning of optogenetics to explore the impact on cooperation and diffusion of public goods in microbial systems.

In this work, we use optogenetics and spatial light patterning to activate the expression of the invertase SUC2 at selected locations within populations of yeast cells. Yeast cells can therefore selectively be switched from cheater to cooperator phenotypes upon light stimulation (Fig. 1b), creating spatially structured landscapes of cooperators and cheater cells (Fig. 1c). Combined with a dedicated experimental system to track the growth of cell populations with time

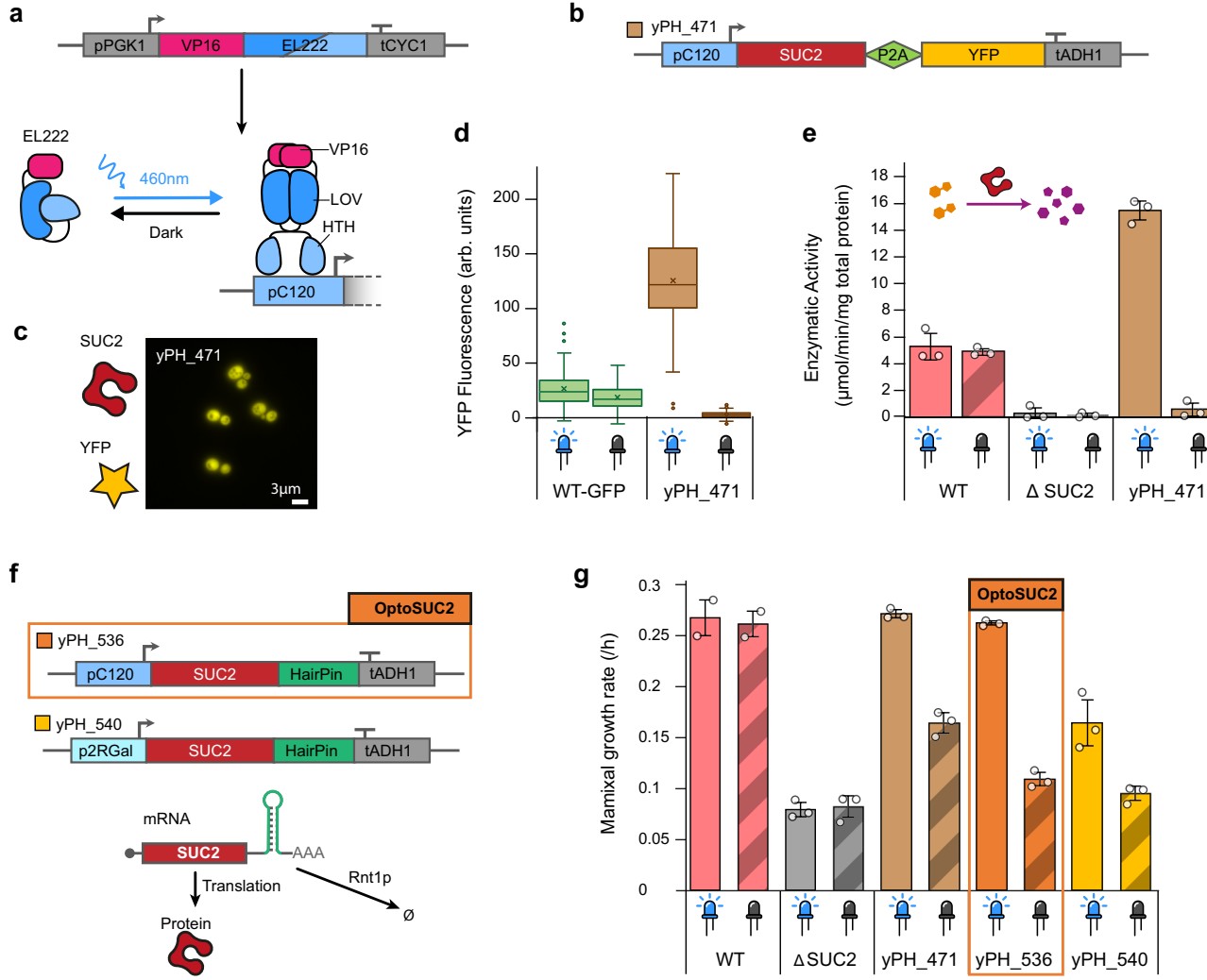

**Fig. 2 | Strain design for tight control of light-induced production of the invertase Suc2p. a** Schematic of the EL222 light-inducible transcription factor and its corresponding specific promoter pC120 to drive gene expression using blue light[39]. **b** Construction of the optogenetic strain with the P2A self-cleaving peptide to separate the fluorescent protein YFP from the Suc2p invertase. **c** Microscopy images of YFP fluorescence in illuminated cells. **d** Fluorescence measurements of the Suc2p-YFP reporter in the presence or absence of light. Cross, lines, boxes, whiskers, and circles represent mean, median, quartiles, 1.5× the interquartile range, and outliers, respectively (outlier exceeds a distance of 1.5× the interquartile range to the closest quartile), with $n = 134, 43, 117, 116$ cells, respectively. **e** Enzymatic activity of Suc2p measured in the dark and upon illumination using a

glucose quantification assay (see Methods). Hexose and sucrose are shown in purple and orange, respectively. Bars represent the means; circles represent technical replicates and error bars represent ±the standard deviation. **f** Improved designs for optogenetically induced expression of Suc2. Reduced mRNA lifetime is accomplished by RNA hairpin-mediated degradation of the transcript (bottom). **g** Maximal growth rate of relevant strains in SC 1% sucrose. The yPH_536 (hereafter called OptoSUC2) exhibited the best dynamic range for inducible growth on sucrose. Bars represent the means; circles represent technical replicates and error bars represent ±the standard deviation over the technical replicates. Source data are provided as a Source Data file.

and a numerical model, we show the existence of two characteristic length-scales of cooperation/competition that involve both cheaters and cooperators and drive the emergence of the spatial landscape within a cooperator/cheater yeast consortium.

## Results

### Light-inducible production of the Suc2p invertase enables extracellular hydrolysis of sucrose

We built a yeast strain in which SUC2 expression was placed under the control of a light-inducible promoter. The strain construct reported by Moreno Morales et al.[29] was based on the CRY2/CIB1 light-inducible system and although it showed interesting properties, we decided to use the more versatile EL222 light-inducible system[33] and performed a set of optimizations to obtain a strain with higher growth rate and induction levels, making it close to the Wild Type behavior when growing on sucrose. We integrated the blue-light sensitive

transcription factor EL222 into the genome of the ΔSUC2 yeast strain under the control of a strong constitutive promoter and the *SUC2* gene under the control of the EL222-dependent pC120 promoter[33] (Figs. 1a and 2a; see Methods). To estimate the expression of SUC2 upon illumination, we fused the gene to a YFP fluorescent reporter with the self-cleaving peptide sequence P2A[34], which ensures that the invertase and the fluorescent reporter are produced as separate proteins[35] (strain yPH_471, Fig. 2b). Indeed, it is known that fusion of invertase to a fluorescent protein can mistarget its extracellular localization[36]. We thus used a P2A peptide which allowed us to have a proxy of gene expression levels while keeping SUC2 functional. Similarly, we built a native SUC2 reporter strain (yPH_484) by fusing P2A-YFP to the *SUC2* gene in the wild-type strain. We measured YFP fluorescence (Fig. 2c, d) and the invertase activity (Fig. 2e) of these strains after 2 h of invertase production under blue light in 24-well plates placed in a homemade light plate apparatus (see Methods and

Supplementary Fig. 2. As expected, the WT and ΔSUC2 strains did not show SUC2 enzymatic activity in response to light, while both light-inducible SUC2 strains exhibited marked increases in invertase activity upon blue light stimulation. Quantitatively, yPH_471 produced up to 310% of the enzymatic activity of WT (under 2.8 mW/cm² illumination).

We then investigated the growth rate of both strains in sucrose upon illumination. We performed these experiments in the same way as the previous experiments: 2 h invertase production in 0.05% glucose, followed by cell growth monitoring in 1% sucrose using a plate-reader (see Methods). First, we observed that the ΔSUC2 strain exhibited slow growth in sucrose, with a maximal growth rate around ~30% of the WT (Fig. 2g). This unexpected residual growth could be due to the presence of maltose symporters, which may have residual activity for sucrose[19]. However, we did not observe any difference when the maltose symporter *MAL11, MAL31*, and *MPH2-3* genes were deleted (Supplementary Fig. 3), ruling out this hypothesis. We therefore attributed the residual growth of ΔSUC2 in sucrose to spontaneous hydrolysis of sucrose, which is known to occur in acidic environments (the pH of media is ~5)[37]. In addition, we observed that— even in the absence of blue light illumination—the light-inducible SUC2 strains grew faster than ΔSUC2. Hence, the basal activity of the pC120 promoter was high enough for cells to produce and progressively accumulate Suc2p, resulting in a significant growth rate of 0.11/h.

The long lifetime of the Suc2p protein (i.e., no loss of activity was measured after 48 h of incubation at 30 °C between pH 4 and 6)[38] likely enhanced the effect of basal leakage from the pC120 promoter and led to the active accumulation of Suc2p in the periplasmic space. Thus, to increase our capacity to control yeast growth based on light-induced invertase production, we optimized our strain construct via two complementary strategies by (1) reducing the leakiness of the pC120 promoter and (2) reducing the lifespan of the *SUC2* mRNA (Fig. 2f). As we previously showed that the yPH_471 strain produced more invertase upon light induction than the levels required to support yeast growth on sucrose (Fig. 2e, g), we did not expect these modifications to drastically reduce growth upon illumination. We thus focused our strain characterization on the growth properties in sucrose, as it is the determinant variable dictating if cells are behaving as effective cooperators. We modified the pC120 promoter[39] by reducing the number of binding domain repeats from five to two. We also added a hairpin mRNA degradation tag[40] in the 3′ untranslated region of the *SUC2* gene. The resulting strains called yPH_536 (hairpin tag) and yPH_540 (changed promoter and hairpin tag), both showed reduced growth rates in the dark compared to yPH_471. When illuminated, yPH_536 had a comparable growth rate to the WT in sucrose, while the maximal growth rate of yPH_540 was 61% of the WT (Fig. 2g). We thus selected the yPH_536 strain, which we call OptoSuc2 in the remainder of this article, since it exhibited the largest contrast in growth rate between dark and illuminated conditions.

### Spatial control of yeast growth can be obtained by light activation of SUC2 in a microfluidic chamber

We first tested the OptoSuc2 strain in a microfluidic chamber perfused with media supplemented with sucrose (Fig. 3 and Methods). At this small scale (i.e., cells are growing as a monolayer in a chamber of 400 μm × 400 μm), we wanted to evaluate how rapidly the hexoses released by a well-defined spatial domain of Suc2p-producing cells diffuse to non-producing cells at the opposite side of the chamber. The cells were grown in the microfluidic system in glucose, starved for one hour, and then switched to 1% sucrose. We used a digital micro-mirror device (DMD) to illuminate a patch of ~400 cells while performing timelapse microscopy (Fig. 3a–c, Supplementary Movie 1). The acquired images were analyzed by particle image velocimetry (PIV; see methods) to obtain a vector field of the cell displacement (Fig. 3d; see Methods) that results from the growth of cells pushing neighboring cells.

Computing the divergence of this vector field gives an estimate of the local cellular growth rate (Fig. 3e, f). Once we illuminated a selected area, we observed cell growth in the same area (Fig. 3d, e, Supplementary Movie 1), demonstrating that cells were indeed producing large enough quantities of hexoses to grow. With time, we also observed that the cheater cells located in the dark region at the opposite side of the microfluidic chamber started to grow but at a slower pace than the hexose-producing cells (Fig. 3e). In comparison, cells trapped in chambers kept in the dark in the control experiments did not exhibit significant growth (Fig. 3f). This suggested that the cheater cells used the hexoses that diffused from the illuminated cells and confirmed that the illuminated cells were acting as cooperators. This experiment demonstrated our capacity to control local growth at a spatial resolution of ~100 μm by patterning a yeast population into domains of induced cooperators and cheater phenotypes.

However, cell growth inexorably pushed cooperating cells (i.e., cells that were expressing the Suc2p invertase) away from the illuminated area (Supplementary Fig. 4 and also Supplementary Fig. 5 showing a similar experiment with a YFP SUC2 reporter−yPH_471). Moreover, because of the long lifetime of Suc2p, the cooperator cells conserved their capacity to produce hexoses even when they were outside of the illuminated area, which effectively blurred the frontier between cooperator and cheater domains and made it difficult to maintain spatial segregation of cooperators and cheaters at such a small scale.

### OptoCube is a device for simultaneous light patterning and microbial growth monitoring on agar plates

Therefore, we next focused on yeast growth on solid media at larger scales (from a few millimeters to several centimeters), which is more consistent with the scale of natural microbial populations such as biofilms. To this end, we built the OptoCube, a temperature-controlled incubator equipped with a DMD to project light patterns on a set of six standard (10 cm diameter) or 15 small (6 cm diameter) agar plates. The agar plates are placed on top of a flatbed scanner to record time-lapse images of microbial growth (Fig. 4a, Supplementary Fig. 6). This setup, which is described in detail in the Supplementary Method 1, allows high spatial and temporal resolution of light patterning (~0.1 mm, ~1 s, Supplementary Fig. 7) compared to the dimensions and dynamics of microbial colonies (>1 mm, > 1 h). We designed a specific protocol to reproducibly obtain a homogeneous lawn of yeast over the surface of the Petri dishes by overlaying a soft-agarose layer (0.67 mm) containing yeast on top of a gel containing sucrose (Fig. 4b). This resulted in a thin (~2.35 mm) and translucent gel that can be imaged using the scanner. Cells embedded in the gel proliferated and formed clumps of microcolonies (Fig. 4c) trapped in the gel, which avoids uncontrolled cell displacement over long distances and/or colony expansion that modify the cheater/cooperator domains, as occurred in the microfluidic device. We calibrated the intensity measured by the scanner (Supplementary Fig. 8) with gels containing known densities of cells and used this calibration curve throughout our experiments to convert the pixel intensities into cell densities in colony-forming units per milliliters (CFU/mL). We also checked that the illumination did not induce significant phototoxicity (Supplementary Fig. 9). As a first demonstration of the capabilities of the OptoCube, we directly projected images onto a lawn of OptoSuc2 cells (yPH_536) growing on top of a 1% sucrose gel (Fig. 4d–g, Supplementary Movie 2).

As expected, growth mostly occurred within the illuminated areas, in which cells were producing Suc2p and were therefore able to hydrolyze sucrose. Embedded yeast colonies grown during our experiments reached size ranging from 10 to 40 μm, small enough so that nutrient diffusion is not limiting within the colony[27,41]. The resolution of the image produced at the surface of the gel was slightly blurred. We hypothesized that blurring occurs due to metabolic cooperation between cells at the frontier between the dark (cheaters)

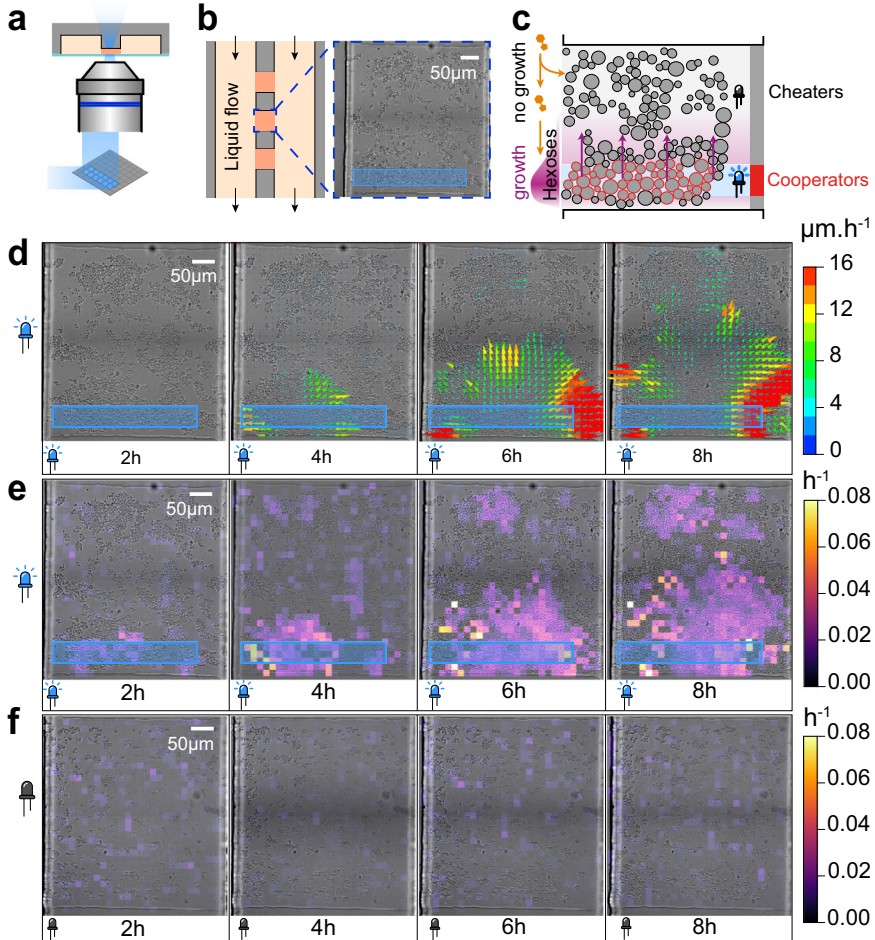

**Fig. 3 | Spatial control of yeast growth in a microfluidic chamber. a** A micro-fluidic chip is used to grow a monolayer of yeast cells in a well-controlled chemical environment through perfusion. A digital micromirror device is used to project a pattern of light onto the field of view. **b** Cells are grown in chambers of typically 400 μm × 400 μm and are perfused via two large channels on the sides of the chambers. The height of the chambers (3.5 μm) ensures that cells grow as a monolayer. A bright-field image of the chamber filled with hundreds of OptoSUC2 yeast cells. Such initial population size and distribution were observed consistently (*n* = 5). The blue rectangle represents the area of the field of view that is illuminated at 460 nm for 200 ms every 6 min. **c** A schematic representation of the diffusion processes occurring in the chamber. Hexose and sucrose are shown in purple and orange, respectively. **d** Displacement vector map obtained through analysis of time-lapse images through PIV (particle image velocimetry). Cell motion only occurs due to cell growth; thus, we expect to observe large vectors (large dis-placement) at the interface between the dark and the illuminated areas (see Sup-plementary Movie 1). **e** Divergence map of the vector field, which is a proxy of the local cell growth rate. **f** Control without illumination, showing no significant growth throughout the experiment.

and illuminated (light-induced cooperators) domains. Indeed, cells in the dark near an illuminated area could grow using the hexoses (public good) produced in an illuminated domain by cooperators.

## Modeling of yeast growth on sucrose

To better understand the roles of the key physical and chemical ingredients involved in this light-induced cooperator/cheater land-scape, we built a simple model of partial differential equations (PDEs) based on the work of Koschwanez et al.[16]. The purpose of this model is to guide the experimental results' interpretation by estimating the spatial variation of sugars concentrations that cannot be easily mea-sured experimentally. This model used: (1) Michaelis–Menten kinetics for enzymatic reactions—i.e., invertase catalysis for hydrolysis of sucrose and high- and low-affinity glucose transporters for yeast con-sumption of hexoses; and (2) a Monod equation for the dependency of the growth rate on the concentration of hexoses. Crucially, the model accounts for the diffusion of the sugars in space. While fructose and glucose utilization by yeast can differ in anaerobic conditions[42], it is unclear what occurs in aerobic conditions and to what extent this would impact the cheating/cooperating dynamics. For the sake of

simplicity, we approximated fructose and glucose utilization as iden-tical (i.e., hexose utilization).

Given the small gel thickness, diffusion equilibrates the con-centrations of the sugars much faster in the vertical dimension than horizontally; thus, we restricted the model to the two horizontal dimensions only. The set of equations and related parameters are described in the Supplementary Method 2. We did not account for other nutrients (notably nitrogen sources) in our model, as we assumed that their availability was not limiting. Importantly, we retrieved most of the parameters (Supplementary Table 1) of the model from the literature and manually tuned only three parameters—the invertase production rates $\alpha_{coop}$ and $\alpha_{cheat}$ and the maximal growth rate $\mu_{max}$—so that the numerical simulations fit both the dynamic and final densities of our experimental observations (light dose response and spatial wavelength experiments). Note that we attributed a small invertase production rate to the cheater cells (in the dark) to account for leakage of the pC120 promoter. We found $\alpha_{coop}$ to equal 1.8E−24 mol s$^{-1}$ cell$^{-1}$, $\alpha_{cheat}$ = 1.5E−25 mol s$^{-1}$ cell$^{-1}$, and $\mu_{max}$ = 0.27 h$^{-1}$. This model was numerically solved using a PDE solver in Python (see Methods).

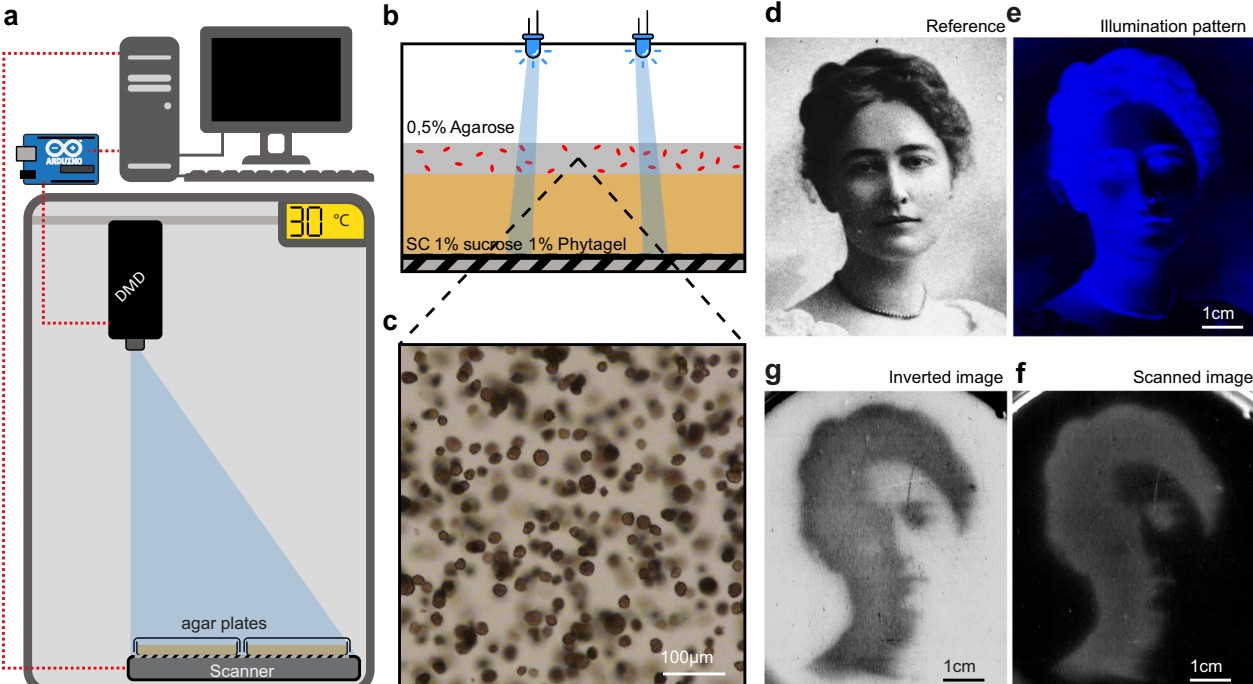

Fig. 4 | **The OptoCube is a home-made device for optogenetic spatial patterning and yeast growth monitoring on solid media at a large scale. a** The OptoCube is composed of a DMD (digital micromirror device) fixed at the top of a temperature-controlled incubator and calibrated to illuminate a scanner placed under an array of Petri dishes. The DMD and the scanner are controlled by a computer and a microcontroller. Under our tested conditions, the light intensities of the DMD pattern on the Petri dishes ranged from 0.0014 mW cm$^{-2}$ to 1.13 mW cm$^{-2}$. **b** Cells were grown in a layer of 0.5% agarose gel on top of a layer of Phytagel containing 1% sucrose and yeast SC media (see Methods). **c** Under these conditions, cells develop into microcolonies with diameters ranging from 10 to 40 μm, consistently observed in all our experiments. In contrast to the microfluidic device (Fig. 2), hexoses and

sucrose can diffuse, and cell growth is constrained within the gel. **d** To illustrate the patterning ability of this device, we projected an image of Maud Menten (courtesy of University Archives, University of Pittsburgh Library System) as a small tribute to her work on the Michaelis-Menten enzymatic kinetic equation, which was developed using invertase as the model[60]. **e** Blue light pattern projected by the DMD on top of a Petri dish containing OptoSuc2 cells for 45 h. **f** Scan of the Petri dish showing the regions where yeast has grown (gray areas). The first image of the timelapse was subtracted as the background. **g** Inverted image of the resulting yeast growth, revealing the image of Maud Menten developed through OptoSUC2-induced cooperation (see also Supplementary Movie 2).

## Varying the light intensity enables quantitative control of the level of invertase and the yeast biomass yield

Next, we investigated how the rate of invertase production influences the growth of yeast on sucrose on agar plates. We thus examined the light-dose response of the OptoSuc2 strain by applying homogeneous and constant light stimulation over the agar plates (Fig. 5, Supplementary Fig. 10). The resulting growth curves (Fig. 5a) were used to extract the maximal growth rates (Fig. 5c) and the final cell densities in the stationary phase (Fig. 5d). By varying the light intensity, we were able to tune the maximal growth rate between 0.12 h$^{-1}$ and 0.38 h$^{-1}$ (Fig. 5c), with the growth rate increasing with the intensity of illumination−as expected. We also observed that the final cell density depended on how fast the cells consumed the sucrose stock (Fig. 5d), with faster growth leading to lower yields. As expected, the simulated growth curves (Fig. 5b) recovered the observed final densities (Fig. 5d) but did not fully fit the experimental growth rate (Fig. 5c) nor the behavior at low light intensities. The gap between experimental data and the model can very likely be improved by explicitly taking into account the dependence of SUC2 expression with light intensity. However, the model sufficiently replicated the experimental observations for high light intensities, which is the conditions we used for our next experiments. Overall, we succeeded in building an experimental system and a mathematical model to quantitatively explore the spatial interactions between cheater/cooperator cells in spatially extended yeast populations. Importantly, using these tools, we can now determine the impact of the relative size of the domains of cheaters and cooperators on their respective growth.

## Emergence of cheater-cooperator and cooperator-cooperator competition depends on the population domain sizes

To start, we projected single lines of blue light with varying widths ($w$) on Petri dishes containing homogeneous lawns of OptoSuc2 cells (Fig. 6a, Supplementary Movie 3). As expected, significant growth occurred in the cooperator domains induced in the illuminated areas. However, growth inhomogeneities became evident when these cooperator domains were wider, with most of the growth occurring in the border zones between cooperator and cheater domains rather than at the center of the cooperator domains. We measured the final cell density profile in the region of interest (at $t = 85$ hours; Fig. 6b) and obtained the mean cell density profile along the horizontal axis. Cell density decayed exponentially at the border between illuminated and non-illuminated areas, confirming that cheaters (in the dark) were growing on the hexose produced by the cooperators (illuminated area) and that the cheater cells were growing more near the frontiers of the cheater/cooperator domains.

To estimate how cooperator and cheater populations share the available sugars, we measured the cell densities at the center of the illuminated area (cooperator domain) and 1 cm away from this area (cheaters) as a function of the width of the illuminated area (Fig. 6c). Interestingly, we observed that increasing the width of the illuminated area decreased the density of cooperators and increased the density of cheaters. The final densities of cooperators were the highest for thin lines of light; this can be explained by the fact that sucrose diffused from the dark (cheaters) domain and supported the growth of cooperators in the illuminated area. This influx of sucrose was mostly used

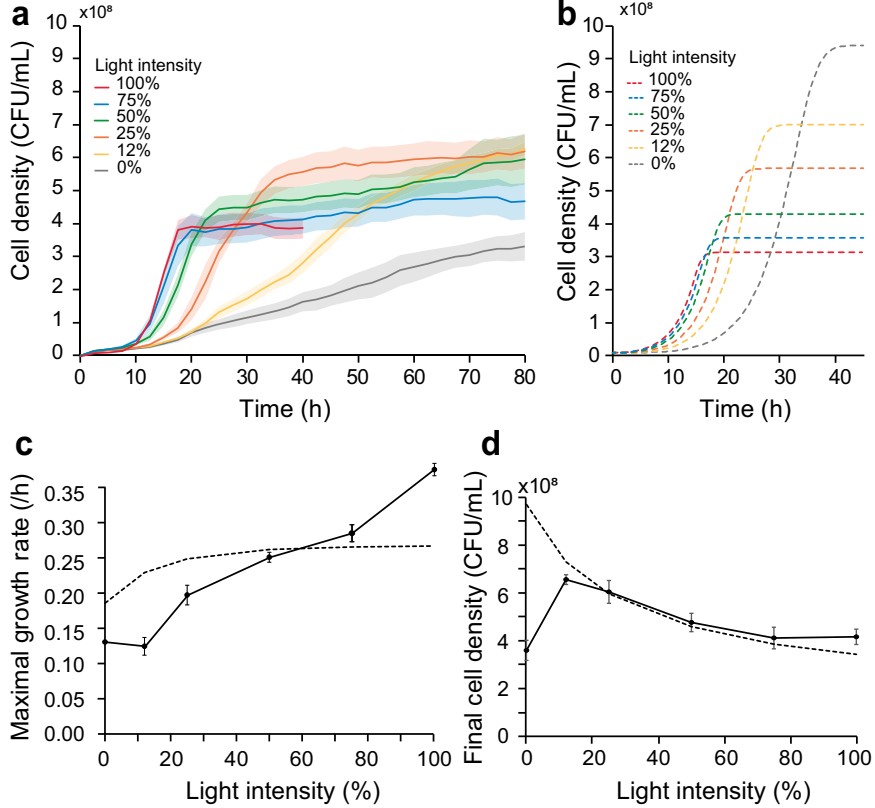

**Fig. 5 | Light-dose response of the OptoSUC2 strain in the OptoCube in response to homogeneous light illumination. a** Experimental measurements of the cell density as a function of time at varied light intensities. **b** Corresponding simulated growth curves obtained with our model. More intense illumination resulted in faster induction of growth and more rapid saturation of cell density. **c** Maximal growth rate determined from the growth curves presented in (**a**). **d** Dependence of the final cell density on the intensity of light. Solid lines represent the mean of the experimental results; dashed lines represent simulated data. The shaded areas and error bars represent ±one standard deviation for three technical replicates. Source data are provided as a Source Data file.

up close to the interface of the dark/illuminated domains and could not reach the center of cooperator areas if the cooperator domains were too large. This led to lower densities at the center of large, illuminated areas and more growth of cooperators (and cheaters) near the frontiers of the domains. This demonstrated that cooperators are competing for sucrose within the illuminated areas, even though they cooperate by sharing another resource (hexoses) as a public good. For large cooperator domains ($w > 5$ mm), this phenomenon led to the highest final cell density profiles at the frontiers of the lines instead of their center (Fig. 6b).

To better understand this process, we simulated one of these experiments ($w = 5.6$ mm wide blue line) and numerically examined the concentration profiles of sucrose and hexose (Fig. 6d) after 15 and 25 h of illumination. These numerical simulations showed that the concentration of hexoses initially increases in illuminated areas, promoting cooperator growth. This is the result of the production and consumption of hexoses in the proximity of its production site. However, the presence of higher numbers of cells that can hydrolyze sucrose depletes sucrose, which in turn reduces the hexose production rate. This mostly occurs for cells that are far from a source of sucrose (*i.e.*, cells at the center of the cooperator domain), which can no longer sustain their growth. The cheaters in the dark domain are unable to metabolize sucrose, thus sucrose continually diffuses from the cheater domain towards the cooperator domain; thus, the cooperator domain can be viewed as a sink for sucrose. Sucrose is primarily hydrolyzed at the frontiers, which promotes the growth of both cooperators (and cheaters through the diffusion of hexoses) at this interface. In other words, the maintenance of cooperator growth (and their cooperation phenotype) is only guaranteed by the near lack of

sucrose uptake by cheaters. Indeed, the cheater domain is a reservoir of sucrose, and its size relates to the amount of sucrose that can feed the cooperators in their own domain. Thus, both the cheater and cooperator domain sizes are needed to explain the landscape of microbial growth.

## Light spatial patterning is processed by cooperators as a spatial bandpass filter that filters out too-small or too-large cooperator domains

Finally, we investigated the behavior of the cooperator/cheater landscapes created using periodic illumination patterns. Cells were illuminated by periodic lines of blue light (cooperator domains) separated by non-illuminated regions (cheater area; Fig. 7a, Supplementary Movie 4). We chose to vary the wavelength of the patterns (corresponding to the sum of the width of a blue line and a dark line) while maintaining a constant light-to-dark area ratio of 25% to 75% to keep the global illumination constant (this fixes the initial frequency of cheaters and cooperators). As a result, all experiments received the same illumination on average, but the width of the cooperator and cheater domains varied. We emphasize the fact that the spatial wavelength of the cooperator/cheater pattern does not correspond to the cooperator population frequency, which is here kept constant at 0.25 across all wavelength experiments through a constant light-to-dark ratio. Here, we are investigating the impact on the spatial organization of cooperators' and cheaters' domains, not that of the ratio of cooperator to cheater cells in a well-mixed culture as it is usually done experimentally.

As previously observed, cells mostly grew in the illuminated areas (cooperator domains); cells also grew in dark areas (cheater domains)

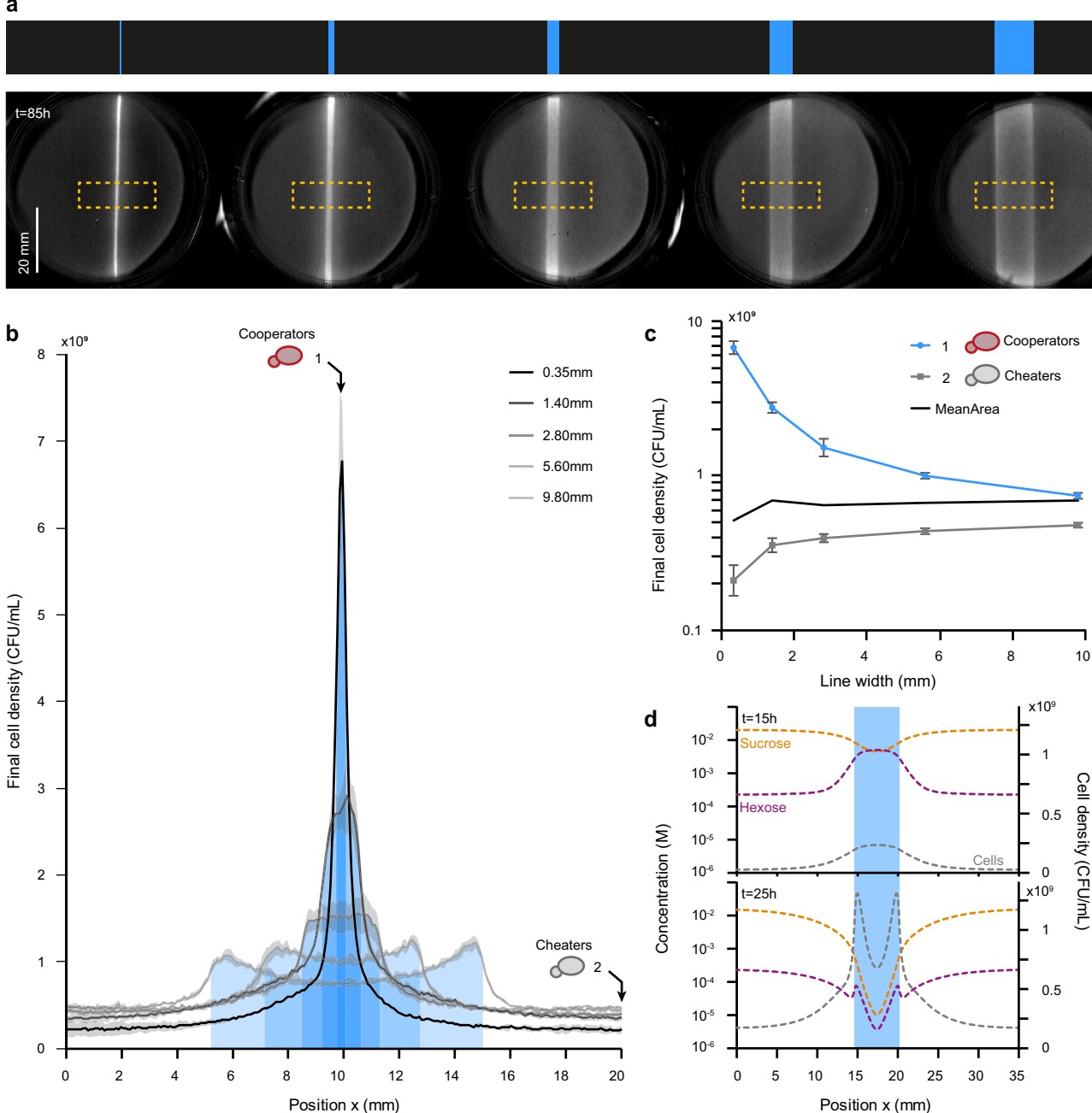

**Fig. 6 | Varying the size of the cooperator domain impacts the density of cooperators. a** Single lines of light of varying widths (top) were projected on cells embedded in a thin layer of gel. The growth of cells in the cooperating (illuminated) and cheating (dark) domains was monitored over time. The bottom images show the cell densities measured by the scanner after 85 h of growth. The background of the first image in the timelapse was subtracted from subsequent images. **b** Cell density profiles (averaged from the dashed orange squares in (**a**) at $t = 85$ h. Increasing the width of the cooperator domain both decreases the final density of cooperators at the center of the line and increases the density of cheaters at the frontier of the dark and illuminated domains. Solid lines represent the means. Shaded grey areas represent ±one standard deviation for three technical replicates. **c** Density of cooperators in the center of the illuminated area ($x = 10$ mm) and density of cheaters far from the line ($x = 20$ mm). The mean final density over the

entire selected area is nearly constant (black trace). Final cell densities are plotted on a logarithmic scale. Error bars represent ±one standard deviation for three technical replicates. **d** Simulated concentration and cell density profiles obtained numerically for a 5.6 mm wide illumination line at $t = 15$ h and $t = 25$ h. Concentrations are plotted on a logarithmic scale. At 15 h, hexoses are produced everywhere in the cooperating domain, which promotes the growth of cooperating cells decreases the sucrose concentration, and creates a source of glucose (public good). This leads to competition for glucose and an increase in the cell density of cheaters located at the frontiers of the cooperating domain. Within the cooperating domain, competition for sucrose leads to a decay in the sucrose concentration towards the center and an increase in the density of cooperating cells at the frontiers with the cheater (dark) domain. Source data are provided as a Source Data file.

but only close to the illuminated domains. In this way, we could determine the dependence of the density of cells across the cheater domains and the cooperating domains as a function of the illumination wavelength. Varying the wavelength provides a way to model the typical patch sizes of populations that can be found in real ecosystems.

Specifically, at small wavelengths, the separation between cheaters and cooperators was blurred. To analyze these observations, we computed the cell densities at the center of the illuminated area (1) and the dark area (2), which represent the final densities of the subpopulations of cooperators and cheaters, respectively (Fig. 7b, c). We

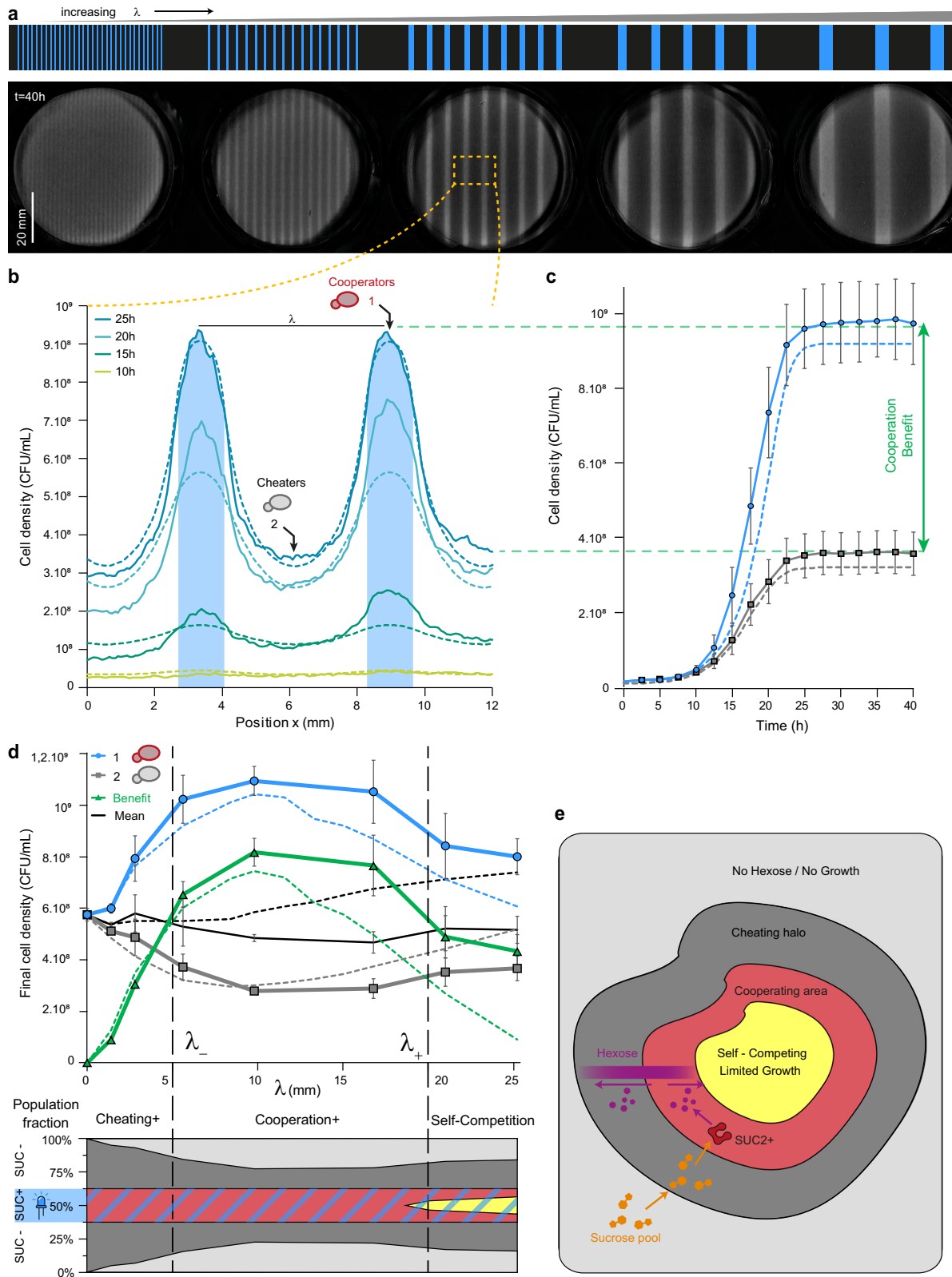

also computed the difference between these two values as a proxy of the cooperator benefit, which indicates how much cooperators grew compared to cheaters. We obtained an asymmetric bell-shaped curve when we plotted the cooperator benefit as a function of the wavelength of the illumination pattern (Fig. 7d).

From the cooperator benefit, we computed two cut-off wavelengths, defined as the distance at which the cooperator benefit is 70%

of the maximal cooperator benefit, namely 5.7E8 CFU mL$^{-1}$, which are attained at ~5 mm (λ−) and ~20 mm (λ+). Below a certain cut-off (<5 mm), the smaller the wavelength, the smaller the cooperator benefit: the cheaters are too close to the cooperators, and as such, the cheaters have access to hexoses under the same conditions as cooperators. Thus, cheaters have the advantage—even though they continue to need cooperators to grow. Conversely, and in agreement with

**Fig. 7 | Periodic patterning of cooperator and cheater domains provides a quantitative measurement of cooperation and competition length scales.** **a** Periodic lines of light of various widths were projected, keeping a constant ratio between dark and illuminated domains (75% dark, 25% illuminated). Cell density was measured after 40 h of growth. **b** Averaged cell density profiles of a selected area (dashed orange rectangle) at different times for the wavelength of 5.6 mm for both experimental (solid lines) and numerical data (dashed lines). This illustrates that cooperators and cheaters grow in their respective domains, but the cooperator cells have a marked fitness advantage. **c** Evolution of cell densities over time in the center of a cooperator domain (illuminated) and the center of a cheater domain (dark). Error bars represent ±the standard deviation for three technical replicates. **d** Maximal cell density as a function of the wavelength of the illumination pattern for cooperators (blue circle), cheaters (grey square), and the averaged population (black solid line). Circles, squares, and triangles represent the means, and error bars represent ±one standard deviation of three technical replicates. Dashed lines

represent numerically simulated data. The green triangles (and lines) correspond to the cooperator benefit, i.e., the difference in cell density between cells in the cooperator domain and cells in the cheater domain. All data were obtained after t = 40 h. The bell shape curve of the cooperator benefit (green triangles) represents a bandpass filter with two cut-off wavelengths. Below the lower cut-off ($\lambda_-$ -5 mm, corresponding to illuminated lines of -1.25 mm), the cooperator domain is too small to retain glucose for its own profit, and cells in the dark areas can grow. Above the larger cutoff ($\lambda_+$ -20 mm, corresponding to illuminated lines of -5 mm), the cooperator domain is too large to be sustainable throughout the domain, given the limited influx of sucrose crossing its frontier. Cells in the center of the cooperating area are now in competition for both sucrose and glucose (self-competition), which decreases the final cell density. **e** Sketch of typical growth domains of cheaters and cooperators for large domains, illustrating that cooperation and growth primarily occur at the frontiers between cooperators and cheaters. Source data are provided as a Source Data file.

our previous observations (Fig. 6b), there is also an upper cut-off (>20 mm), above which the cooperator benefit decreases as the wavelength increases further.

The upper cut-off can be explained by self-competition between cooperators for sucrose. Indeed, sucrose diffuses from dark areas to illuminated areas and cannot reach the center of the cooperator area if the cooperator area is too large. More generally, we can interpret these results using an analogy with spatial filters: the system behaves as a bandpass filter, whose transfer function (the cooperator benefit), has a low cut-off wavelength ($\lambda_-$ -5 mm) due to cheater-cooperator competition for hexoses and a high cut-off wavelength ($\lambda_+$ -20 mm) due to cooperator competition for sucrose (and hexose). Using our numerical model, we could further check that each cut-off wavelength correlates to the diffusion of the corresponding sugars (low hexoses; high sucrose) by artificially tuning the diffusion coefficients of these nutrients independently. Indeed, from the set of equations of our model, we expected the cut-off wavelengths to increase when the corresponding sugar diffusivity increases and is what we obtained numerically (Supplementary Fig. 11). Therefore, our results suggest that the diffusion of sucrose (a reserved carbon source for only cooperators) and hexose (a public good for every cell) define the cut-off dimensions of the bandpass filter, however, further experiments are needed to quantitatively test this hypothesis. Taken together, the data in this study demonstrate that the size of the cheater and cooperator domains is a key determinant of the cooperators' benefit and population growth.

## Discussion

In this study, we built a yeast strain that produces the sucrose invertase Suc2p upon illumination with light and tested the population growth dynamics and the benefits of cooperation in various artificially shaped landscapes of cooperators and cheaters. The OptoSuc2 strain indeed acts as a cooperating cell when illuminated but remains a cheater when kept in the dark because non-illuminated cells cannot metabolize sucrose and instead rely on the hexose produced by adjacent cooperating cells. We were, therefore, able to quantitatively explore the spatial metabolic interactions between cooperators and cheaters.

Our main finding is the existence of two typical length scales that set the domain size of both cooperators and cheaters. Both length scales are defined by the diffusion and uptake properties of hexose and sucrose. The first length scale is that of competition between cheaters and cooperators and can easily be understood as the typical length over which hexose diffuses away from cooperators. As exemplified in Fig. 7e, if cooperator domains are smaller than this wavelength, the number of cheaters that benefit from the produced hexose is comparable to the number of cooperators, and cooperating provides no clear benefit. In other words, cooperator domains need to be larger than $\lambda_-$ to be distinguishable from cheater cells: too-small domains are equivalent to what would be obtained with a homogenous mixture of cooperators and cheaters.

The existence of the second upper length scale, $\lambda_+$, was unexpected and demonstrates the benefit of grouping cooperators decreases when their domain is too large. We attribute this decrease to the fact that cooperators not only interact by producing hexose that benefits their neighbors (cooperation) but also by competing for the basal carbon source, in this case, sucrose. This competition means that cells far away from the sucrose source obtain less sucrose and, as such, produce less hexose. This is similar to the growth dynamics observed in any extending colony, for which growth occurs mostly at the edge of the colony where nutrients are abundant. Therefore, in a spatially structured cheater/cooperator system, the existence of large domains of cheater cells (which cannot hydrolyze sucrose) ensures the presence of secured pools of sucrose that can diffuse toward cooperator islands and be used first by the cooperator cells located at the frontier between cheater and cooperator domains. Competition for sucrose takes place within a cooperator domain, and the cells closest to the cheater domains are at an advantage. Said differently, cooperating cells benefit from proximity to cheater cells, and cheater cells not only function as cheaters but also as key actors that facilitate the growth of cooperating cells at the domain frontiers. Therefore, cheater cells also help the cooperators to grow faster in the vicinity of the cheaters' domain of existence. Furthermore, as exemplified by our study, this beneficial role of cheater cells is only apparent when the domains of both cheaters and cooperators are large enough.

As we proposed in the Results, this relationship can be summarized by an analogy with a spatial bandpass filter, in which critical wavelengths are linked to the typical distances of the metabolic interactions in the SUC2 yeast system: a lower cut-off wavelength ($\lambda_-$ -5 mm) due to cheater-cooperator competition for hexoses and a higher cut-off wavelength ($\lambda_+$ -20 mm) due to cooperator self-competition for sucrose. We confirmed this analysis with our numerical model, which even though it is only a minimal model with several limitations (e.g., it does not capture well the dependency with light intensity), is able to reproduce the bandpass filter behavior, indicating that it has the main ingredients to explore spatial interactions between cooperators and cheaters. Thus, thanks to the ability to artificially create cooperator/cheater landscapes with light, we defined the optimal range of domain sizes that create cooperating microbial niches. We anticipate that our approach could be applied to other microbial ecosystems to explore the parameters that define the landscape of metabolic interactions.

Our study illustrates the power of optogenetics and spatial patterning to decipher the metabolic interactions at play in spatially complex multicellular assemblies such as colonies, biofilms, and engineered consortia. There is a growing interest in engineering microbial consortia[43–45], in which different types of cells cooperate to more efficiently achieve specific biological functions (bioproduction[46], living materials[47], or live therapeutics[48,49]). Thus, it is essential to better grasp the physical limitations of such systems−in particular, the

impacts of chemical diffusion, the composition of consortia, and their metabolic interdependence—on the dimensions of microbial niches in such applications. We anticipate that optogenetics could be used to locally change the cellular metabolic capabilities of microbial consortia by controlling the size of the domains of the species. Such experiments will help to better understand cooperation and competition mechanisms in microbial ecosystems and how to control complex synthetic microbial consortia in real-time. Importantly, we showed that intrinsic dimensions exist for microbial niches and can be played with to optimize cheating and/or self-competition for resources. This study can guide synthetic biologists to appropriately set the dimensions of engineered living materials[50,51] (ELM) in microbial niches to sizes that are compatible with the desired properties of the ELM, which is a crucial step required to obtain precise functionalities and efficient external control. We extrapolate that such spatial constraints could also be considered when studying the spatial organization of imbalanced microbiomes (dysbiosis), which are linked with major problems such as human obesity, diabetes, skin disease, and a myriad of other diseases due to alterations in the human gut microbiome[52], or unsustainable farm soil fertility associated with a high need for nitrogen fertilization[53].

## Methods

### Yeast strain construction
All yeast strains used in this study are derived from the BY4741 yeast background (EUROSCARF Y00000) and are listed in Supplementary Table 2. All strains have the nuclear marker mApple-HTB2 and an EL222 expression cassette. The marker and cassette were integrated using a classic lithium acetate transformation protocol at the HTB2 or HIS3 locus using kanamycin G418 resistance and histidine auxotrophy, respectively (using plasmids pPH_330 and pPH_297, respectively). All other genetic modifications were undertaken using the CRISPR/Cas9 system[54]. Guide RNA sequences (gRNA) were obtained from oligo synthesis (IDT) and integrated into the plasmid pML104, which already possesses a *Cas9* expression cassette and URA3 marker for auxotrophic selection. The repair strands were obtained from either oligo hybridization for deletion or were PCR-amplified from appropriate plasmids for integration (Supplementary Table 3).

### Plasmid construction
Plasmids were built using a custom Modular Cloning (MoClo)[55] approach, in which standard genetic parts are assembled via GoldenGate assembly. The YTK kit (Addgene; Kit #1000000061) was used as a source for all relevant non-coding DNA sequences—including promoters and terminators, as well as pre-assembled integration vectors—and was supplemented with our own components: the pC120 promoter, the SUC2 protein, the self-cleaving peptide P2A, the minimal promoter p2RGal, and the mRNA HairPin degron tag.

### Growth conditions
Cultures (2 mL) were grown overnight (-18 h) in 14 mL culture tubes (Falcon) in YPD media. Day cultures (5 mL) were grown to exponential phase in filtered synthetic complete media (SC) supplemented with 2% glucose. All cultures were incubated in an Innova 4230 incubator at 30 °C with orbital shaking at 250 RPM. SC media is composed of 6.7 g Yeast Nitrogen Base without amino acids (Difco 291940) and 0.8 g complete supplement mixture drop-out (Formedium DCS0019) in 1 L. Care was taken to reduce unwanted light exposure as much as possible before the start of the experiments: thawed strains on agar plates and precultures were covered in aluminum foil, and all cell handling procedures were conducted without direct exposure to light. To prevent uncontrolled hydrolysis of sucrose in water, we used 20% *w/v* sucrose stock solution buffered at pH = 8 with 1 mM Tris buffer and stored the solutions at 4 °C.

### Enzymatic quantification
We measured the fluorescence and invertase activity of cells after a 2 h enzyme production phase. To compare the expression of invertase in the synthetic strains and WT strains, the production phase was performed in a 0.05% glucose liquid media to repress the native SUC2 promoter. Overnight cultures were washed with 10 mL of sterile water, resuspended in 10 mL of SC media at 0.05% glucose to obtain a final $OD_{660}$ of 0.5, then 1 mL aliquots were placed in triplicate in plastic-bottomed black 24-well plates (Eppendorf Cell Imaging Plates ref. 0030741005), and the plates were placed onto a custom made light plate apparatus (LPA)[56] to control blue light illumination in each well ($\lambda$ = 460 nm, I = 2.8 mW/cm$^2$). After 2 h of illumination at 30 °C with orbital shaking (at 225 RPM), 500 μL aliquots of the cultures were removed, placed on ice to inhibit yeast growth, and then washed with 500 μL of 10 mM sodium azide solution to block further glucose import. The aliquots were then washed with 500 μL of 50 mM sodium acetate buffer (pH = 5.1) and resuspended in 300 μL of the buffer; 100 μL was placed into a PCR tube for the enzymatic activity assay and 200 μL was incubated on ice for the Bradford protein quantification assay.

For the enzymatic activity, 10 μL of fresh 1 M sucrose solution was simultaneously added to all samples using a multichannel pipette, and the reactions were incubated at 37 °C for 10 min in a thermocycler and then heated to 99 °C for 3 min to denature the invertase. Glucose concentrations were determined using the colorimetric enzymatic Glucose (HK) Assay Kit (Sigma) and a Cary 50 Scan spectrophotometer.

To determine the corresponding total protein contents for normalization, glass beads were added to the 200 μL aliquots and the cells were lysed by three rounds of 3 min vortexing separated by 1 min incubation on ice to prevent overheating. The supernatants were serially diluted with water and 100 μL samples were mixed with 100 μL of Bradford assay solution in 96-well plates. The plate was placed in an EnSpire plate reader (PerkinElmer), and incubated for 5 min at 25 °C with mild shaking, and the absorbance values were measured at 595 nm.

### Microscopy
Cells were placed on an agar pad on a glass slide and covered with a coverslip for microscopic observation. Fluorescence images for quantification were captured using an inverted Olympus IX81 epifluorescent microscope, an Xcite exacte light source (I = 30%), and a filter cube (EX) 514 nm/10 (EM) 545 nm/40 (49905−ET) with a 1000 ms exposure time. Yeast cells were segmented using Yeast-spotter[57]. To image cell colonies embedded in the gel, we used a Nikon AZ100 macroscope equipped with a Nikon camera Color DS-Ri1 and a 2x objective AZ-Plan Apo (NA: 0.2/WD: 45 mm).

### Growth rate quantification
After the 2 h invertase production phase described above, we added 25 μL of 20% w/v sucrose stock solution (1 mM Tris buffer pH = 8) to obtain a final sucrose concentration of 1%. The plate was placed in a Spark plate reader (Tecan) at 30 °C and the OD was measured at 660 nm every 5 min (average of five measurements at different locations within one well). In between these measurements, cells were kept in suspension by 240 s of orbital shaking with an amplitude of 4.5 mm followed by 60 s of linear shaking with an amplitude of 6 mm. The resulting growth curves were fitted using the function smooth spline in R with 15 degrees of freedom. The derivative of the fitted curves was obtained, and its maximum value was used as the maximal growth rate per well.

### Microscopy imaging in microfluidic chambers
We used an automated inverted epifluorescence microscope Olympus IX83 equipped with a Motorized stage (Prior Pro scan III), a Zyla

4.2 sCMOS Andor camera with an Olympus 20× Plan (N.A. 0.4) objective, a DMD (MOSAIC3 Andor) and a pE-4000 CoolLed as a light source to achieve spatially resolved optogenetic activation. For optogenetic activation, we used a 460 nm LED at 20% intensity through a filter cube (EX) 470 nm/40; (EM) 525 nm/50 (49002) with a 200 ms exposure time every 6 min. Brightfield images were also captured every 6 min. Microscopy experiments were carried out in a thermostat chamber set to 30 °C. Liquid perfusion was delivered using an Ismatec IPC (ISM932D) peristaltic pump at 50 μL/min. Particle image velocimetry (PIV) analysis was performed using the open-source JPIV software (https://github.com/eguvep/jpiv/) for an interrogation window containing 128 pixels. Isolated non-significant vectors were removed from the velocity vector map. We retrieved the growth rate map by determining the divergence of the velocity field. We did not display the negative values in the divergence map as negative values are due to unchanged pixels in the areas that do not contain cells at the border of the microcolonies.

### OptoCube

The OptoCube is composed of a static incubator (Memmert UF160) equipped with a prototyping DMD projector for light stimulation (DLP® LightCrafter™ 4500 TexasInstrument) and a flatbed scanner (Canon LiDE400) for image acquisition. VueScan© software running on a Windows computer is used to control the scanner to perform periodic scans to measure cell growth in the Petri dishes over time. The DMD pattern sequence is controlled by a microcontroller board (Arduino Uno) that is synchronized with the image acquisition. This sequence can stimulate and monitor up to 15 Petri dishes (6 cm diameter) per experiment. The light intensities of the DMD pattern range can be adjusted from 0.0014 mW/cm$^2$ to 1.13 mW/cm$^2$ using a blue LED at $\lambda = 460$ nm. Intensity measurements were conducted using a power meter (TOR Labs PM100D with S120C sensor), placing the sensor at the same position as where the cells would have been growing during the experiment. The incubator was set to 50% internal ventilation and 50% air inlet. A calibration curve (Supplementary Fig. 7) was made to convert the scanner data into cell density. More information on how to build and use the OptoCube is provided in the Supplementary Method 1. For the data analysis, only the central part of the plate was assessed to avoid boundary effects, and background subtraction was performed using the first image in the timelapse.

### Plating procedure

Bilayered agar plates were prepared before each experiment. The bottom layer was composed of 3.8 mL SC media with 1% w/v Phytagel and 1% w/v sucrose. To avoid hydrolysis and caramelization of the sucrose, the SC media and Phytagel solution was autoclaved, then buffered 20% w/v sucrose stock solution (1 mM Tris buffer pH = 8) warmed to room temperature was added to the hot gel. The solution was poured into small Petri dishes (60 mm; TPP ref. 93060) using 5 mL serological pipettes under a laminar flow-hood (Thermo Fisher MSC-Advantage™ 0,9) and allowed to solidify for 5–10 min. The top layer was made of 0.5% w/v agarose D5 (Euromedex ref. D5-C) mixed with a concentrated solution of washed cells at a 40:1 ratio to obtain a final OD of 0.1 in the agarose gel. Briefly, the agarose was melted in a microwave, aliquoted into 15 mL Falcon tubes, and equilibrated at 46 °C in a water bath, then the cell solution was added and 1.5 mL aliquots were immediately plated on top of the bottom layer using a 5 mL serological pipette. The plates were allowed to solidify for 5 min, then the lids were added, and the plates were sealed with Parafilm.

To avoid issues due to light reflection from the scanner, we held the lids at a 5° angle using a homemade 3D-printed adapter. To reduce droplet formation due to condensation, the inside of the lids were washed with 1 mL of 0.05% (v/v) Triton 100× in 20% ethanol. For the image patterning (Fig. 4e–g), we used a standard-sized Petri dish (external diameter 94 mm Greiner ref. 633180) with 10 mL of SC media containing 1% w/v Phytagel and 1% w/v sucrose overlayed with 4 mL of agarose cell suspension. All plates had a total gel thickness of 2.35 mm (1.68 mm nutritive layer and 0.67 mm yeast growing layer).

### Simulations of yeast growth on sucrose

All simulations were performed by numerically solving a set of partial differential equations. The model and its parameters are detailed in the Supplementary Method 2. The numerical solution was obtained in Python using the PDE solver scikit-finite-diff package[58].

### Reporting summary

Further information on research design is available in the Nature Portfolio Reporting Summary linked to this article.

## Data availability

All materials used in this study, including plasmids and strains, are available upon request. Data used to generate the figures and analysis of this article are available in a public Zenodo repository[59]. Source data are provided in this paper.

## Code availability

The code used for the numerical simulations is available at Github. The code used to pilot the optoCube is also available at Github.

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

## Acknowledgements

The authors would like to thank J. Avalos (Princeton) for kindly providing the original EL222 plasmids, as well as the Nikon platform of the PICT IBISA imaging platform at the Institut Curie. The authors thank their colleagues for reading this paper critically. P.H., M.L.B., A.B., S.P., C.C. were supported by the European Research Council grant SmartCells (724813) and from ANR grants ANR-16-CE12-0025-01 (PH, SP), ANR ANR-16-CE33-0018 (M.L.B., P.H.), ANR-11-LABX-0038 (P.H., B.S., V.S.), and

ANR-10-IDEX-0001-02 (PH). S.B. is supported by a PhD fellowship from ITMO Cancer (France).

## Author contributions

M.L.B. performed all experiments. S.P., A.B., C.C., and S.B. contributed to molecular biology design and construction. M.L.B. and V.S. worked on the mathematical and numerical models. M.L.B., A.B., and P.H. conceived and designed the study. M.L.B., B.S., A.B., and P.H. wrote the paper.

## Competing interests

The authors declare no competing interests.
