## [Peer Review File · Nature Communications]

Optogenetic spatial patterning of cooperation in yeast populationsReviewers' Comments:

Reviewer #1:

Remarks to the Author:

Comments to the Authors:

In this study, the authors use optogenetic control of the invertase enzyme Suc2 in populations of budding yeast distributed in roughly two dimensions to show that glucose diffusion and sucrose diffusion rates determine "cooperator benefit". Namely, when cooperators and cheaters are distributed at different spatial scales, how much cooperators benefit from their ability to make invertase (Suc2). The authors engineer optogenetic control of expression of Suc2 in budding yeast using the EL222 optogenetic system as well as perform some tuning with an mRNA degradation tag and promoter engineering to identify a variant with the highest dynamic range under light control. There is some nice synthetic biology here, and the authors explain their logic and characterization of the system well. However, it is worth noting (also mentioned below) that Moreno Morales, et al already made a functional, optogenetically controlled Suc2 in budding yeast and the authors, in my opinion, really gloss over this point. The current study's system may have some benefits over the previously engineered system, but at least make clear that this approach has already been implemented in the literature in the same organism. The authors then use an OptoCube to spatially designate cooperator and cheater populations using patterned light on petri dishes. They vary the size of these populations (this reduces to a 1-dimensional problem due to the thinness of the agar cells are seeded in and the length of the strips such that only width "matters") and identify a short length-scale cutoff presumably due to glucose diffusion rates and a longer-range cutoff due to sucrose diffusion. In between cooperators receive maximal benefit in the presence of cheaters.

This study employs elegant experimental techniques that have not yet been widely applied to study microbial interactions. The choice of invertase expression in budding yeast is a good one, as it is a well-studied system with vast literature. The main finding, of a cooperator benefit bandpass, is interesting but expected in the context of diffusion rates. The study needs to address major comments before it is appropriate for publication:

Major Comments:

- The authors state (Line 74) that a method to quantitatively control spatial organization of cooperators and cheaters in a population of microbial cells would enable understanding of how populations of cells utilize shared resources. This is very true. However, the authors gloss over the fact that at least two prior studies have demonstrated this approach. Indeed, that is exactly what is done in Moreno Morales (Ref. 23) using an almost identical approach to the current study, namely optogenetic control of Suc2 expression. While it is true that the current study pushes the spatial study much further and uses a significantly refined technique (namely the OptoCube) for spatial control both spatial control (and patterning) were demonstrated in Moreno Morales, et al. The authors also fail to mention that optogenetic, spatial control of a public good (matrix production) was first demonstrated in the bacterium *Sinorhizobium meliloti* (<https://doi.org/10.1021/acssynbio.0c00498>).
- The authors do not directly demonstrate that sucrose and glucose diffusion rates are controlling the length scale cutoffs of the cooperator bandpass filter. While the evidence, when combined with the computational model, is certainly suggestive it would be more compelling if they could change diffusion rates using different concentrations of agar, for example. I also feel like there might be a compelling analytical connection here.
- Why do the authors show data using yPH_470 (which has invertase fused to YFP?). It is known from previous work that fusion to invertase disrupts its secretion and in fact, I would image the authors see some growth defects. (Protein accumulation does seem to be apparent in Figure 2C, although it is difficult at this magnification to tell exactly what is happening). As the authors do mention, this is almost certainly the source of difference in function between strain 470 and strain 471.
- The data in Figure 3 seems disconnected from the rest of the paper. While these experiments are visually appealing, I don't think that they reveal any new or quantitative information. If you excite a clump of cells to produce invertase, eventually cheaters will be able to grow at a distance as hexose diffuses. But this isn't quantified or used to gain new insight. Also, while the caption states that the

DMD can illuminate single cells, this functionality does not seem to be used. In addition, the authors state that within the device a “few cells” are excited with light to produce invertase. But in fact, looking at the figure it looks like at least 10’s of cells, and maybe 100’s. (Line 161)

- In Figure 4 (C) would it be possible to estimate the actual range of colony diameters? The authors state 10’s of microns, but looking at the figure I would say some are on the order of 100’s of microns. It is hard to tell from the figure. Also, a more interesting question: Do the sizes of these cell clumps affect the spatial dynamics? This is a spatial scale at play in the system, and it is known from Koschwanez, et al to have an effect, but does it affect the authors’ results? Does the seeding matter? IE if you seed less densely presumably the cell blobs can grow larger than if you seed more densely and there is more competition between cooperators?
- Modelling: The model fits between Figure 5A and Figure 5B are quite different. Can the authors elaborate on what they think is missing in the model to generate this systematic error at low light intensities? It also isn’t clear that you see saturation in maximal growth rate in the experiments as is seen in the model? (Perhaps difficult to go higher experimentally). In the model, where is there only one cell density term (d_{Cell})—doesn’t there need to be at least two (cooperators and cheaters) as d_{Cell} determines invertase concentration (E)? Also is it clear why dE/dt should depend on hexose concentration? We know that at low glucose concentrations, cells produce a significant amount of invertase.
- In Figure 6C, why are the areas characterized at $x=10mm$ and $x=20mm$. For the cheaters, does it make more sense to characterize their density at a constant distance from the cooperator “edge”? Also looking at the modeling in Figure 6D there seems to be a significant amount of hexose accumulating away from the cooperator domains. Why don’t we see more cheater growth? Are the models and experiments run to steady state? Or is the timing of image acquisition quite important for determining these effects? How stable is the cooperator benefit over time? Would it be reduced if cheaters were allowed to grow longer to take advantage of the accumulating hexose?
- How the diffusion constants affect the cutoffs is not well described in the main text. There is some additional information and modeling in the supplemental text (S10). But are the bandwidths defined by eyeballing arbitrary cutoffs? Is there a relationship between the diffusion constant and the length scale that is more analytical?
- It is interesting that the averaged population (cooperators + cheaters) is constant across the whole range of light patterns. This conflicts with some work from the Gudelj research group which shows that mixes of cooperators and cheaters are actually more efficient. In fact, reference to the Gudelj lab work, which is extremely relevant to the questions outlined in this study, is conspicuously missing. The authors would do well to situate their current findings with their relevance to this previous body of literature.
- Should the longer length scale depend on cheaters at all? How would this change if there were stripes with no background cheater cells (either with modeling or experimentally—which I realize would be more difficult to do in this system). The discussion casts the upper length scale as having to do with cooperator-cheater interactions, but isn’t it really just about the width of the cooperator islands? In general, the no-cheater control is missing from most experiments.
- Please provide more details on the model fitting. The Methods say manual tuning and no fit code is provided in the GitHub code. Was additional model validation done? Was there a loss function used at all?
- It isn’t clear from the supplemental how the issue of light leakage from the DMD with the mirrors “OFF” was solved? It is mentioned as a problem, but what was the solution? And what effect, if any, does this have on your measurements and results if cells are getting some small level of constant illumination?
- There is so much theoretical and experimental literature on the invertase regulation system in yeast and its implications for cooperativity in microbial populations. The discussion would benefit from tying the current results to this body of work.

Minor Comments:

- It is not clear from the references or methods what 2A peptide was used. This can affect efficiency greatly (although apparently the efficiency in the authors’ case based on function is quite good).

- A word of caution (Line 287): using wavelength for the width of the illuminated areas might be misconstrued as meaning a different frequency of light.
- What is being measured by the scanner is not clear. I am assuming it is just optical density, and that the fluorescence of the cells is not used past the first few Figures. Would be nice to see where the scanner's capabilities start to saturate as a function of density---what is the range that you are able to measure? And how does this relate to how densely cells grow on the plate? As this is presumably initial seeded densities. Especially since some of the final cell densities seem to be measured at 10^9 (ie Figure 7)—is this within the linear range of the scanner?
- What is low ambient light in the Methods? Is this low light enough to excite EL222 (for some proteins, ie, cryptochromes, it absolutely would be, EL222 may not be as sensitive)
- Should you convert Go of RAM into GB of RAM (under "Simulation", ie French vs English equivalents)? It isn't that important.
- Can you put a scale bar on Supplementary Figure 8—are these single cells that I am seeing, or clumps of cells? (I'm guessing clumps of cells assuming Figure B is on the same scale as subfigure A?)
- The legend of Supp Figure 9 is swapped in terms of which (A or B) is the dark experiment and which is the light experiment.
- Multiple subfigures labels in Supplemental Figure S10 are not consistent with the number of caption descriptions.

Reviewer #2:

Remarks to the Author:

Spatial organisation of microbial communities is a critical aspect in microbial communities' ecology and its understanding can be extremely useful for microbial community engineering and modelling. In this work Le Bec et al., first engineered an optogenetic system that regulates the breakdown of sucrose in the budding yeast *S. cerevisiae* upon light stimulation. They optimise the optogenetic regulation such to diminish the leaky invertase production and be able to fine tune its expression. Then they tested their optogenetic system, first, on a small-scale, using a microfluidic device where they show that spatial organisation induced by their system can rapidly deteriorate upon preservation of Suc2p activity in the producer. Then, they developed a system called OptoCube to regulate the optogenetic system activation on a larger scale in petri dish. They demonstrated how carefully tuning of this system can help in understanding and predicting growth of cheater and producer sectors in a controlled setup and showed that both cooperation and competition in the producer sectors can emerge depending on the spatial structure used.

I think the work is sound and well executed. I only have few minor comments

- Line 136, I think that the enzymatic activity of invertase has not been screened in yPH_536 and yPH_540 to evaluate its own activity in light and in the dark as has been done with the other strains. Here the authors are referring to maximum difference in growth rate rather than invertase production, which may be expected but not shown.
- Line 158, should the cells that have been kept in the dark still exhibiting a low growth rate as indicated in Figure 1G?
- In supplementary Figure S4 the authors refer to the strain yPH_471 in the figure caption but in the main text the paragraph describes the behaviour in the microfluidic device of OptoSuc2 (yPH_536). To which strain is the Supplementary figure referring to?
- In Figure 5, 0% and 12% light intensity seem to behave differently in the experimental data compared to the simulated ones. What are the reasons that may explain this different behaviour at low illumination intensity?
- Figure 6D, what happens if the simulation is run for the same amount of time of the experiment on plate?
- I could not find the GitHub page for the OptoCube proposed in the supplementary material at the following link https://github.com/Lab513/DIY_OptoCube
- Plasmids sequence and primer sequences should be released with the paper.

- Supplementary Figure S9 the figure caption called Figure A and B in the reverse order. A should be the illuminated one and B the growth curves obtained in the dark.
- Line 319 and line 178 please use the same format for referring to the supplementary figure as done elsewhere.
- Legend Figure 3: If I understood correctly the setup of the system in the legend of figure B and C the chambers should be of typically $400\ \mu\text{m}$ and not $400\ \mu\text{m}^2$ as reported.
- In Supplementary Figure S10 the description of panel C, D, E, F is missing.
- I wonder if there is a specific reason why in Figure 3D the displacement vector map presents a high displacement mainly on the right side of the chamber although the illuminated area encompass a wider part of the device in particular the central part. Is it due to the closeness of the microfluid device side to the illuminated area or is it just a stochastic effect?
- In the simulation paragraph, please change the unit of the RAM used to Gb instead of Go.

Reviewer #3:

Remarks to the Author:

Please see attachment.

Optogenetic spatial patterning of cooperation in yeast populations.

Matthias Le Bec¹, Sylvain Pouzet¹, Céline Cordier¹, Simon Barral¹, Vittore Scolari^{1,2 3}, Benoit Sorre¹, Alvaro Banderas¹, Pascal Hersen¹, * 4

The manuscript describes an interesting method to investigate metabolic interactions between “producer” and “cheater” yeast cells using optogenetic control. What differentiates this study from other similar works studying competition and cooperation mechanisms is that the developed optogenetic quantitative method provides a way to create spatially structured populations with fine control to study behavior and dynamics at the optimal granularity. Combined with an imaging system capturing and tracking cell growth and the developed computational method the authors are able to measure systematically the cost efficiency of the production of a diffusible public good in the cooperating cells and its use by the cheater cells.

The authors carefully constructed the yeast system, tested the functionality with a YFP reporter, optimized optogenetic expression of target protein with careful control of expression dynamics (mRNA degradation hairpin). The authors successfully determined optimal band size for producers and cheaters to maximize growth. The study is complete with a model that describes well the observed patterns. The experiments are well designed, and the authors describe the steps taken to design the optimized setup that was used to come up with the final conclusions. The methods are well described with sufficient details for reproducibility, the data available in a public repository (Zenodo), and the diffusion model is accessible on GitHub in a Jupyter Notebook format. This study helps in future endeavors for optimal engineering of designer microbial consortia via optogenetic regulation in biomanufacturing and other areas.

In a follow-up study it would be interesting to see collaborator/cheater behavior when circular illumination patterns are used.

The reviewer recommends this manuscript for publication with minor revisions.

1. The authors combine glucose and fructose into hexoses in the modeling and explanation of the experimental outcome. There is a difference between glucose and fructose utilization by yeast under anaerobic conditions. There are many papers describing the preferential use of glucose over fructose, which is especially important in wine making (undesired sweetness of the residual fructose in dry wine). <https://doi.org/10.1016/j.femsyr.2004.02.005> The literature is lacking comparative studies under aerobic conditions. Some preference for glucose was identified in aerobic bioreactor runs, but this seems strain specific (<https://doi.org/10.1093/femsyr/foab021>).

2. It is worth noting that the SUC2 locus encodes two different forms of invertase, the glycosylated homodimer excreted into the periplasm and a cytosolic form. The cytosolic form lacks the signal peptide due to alternative translation. The periplasmic enzyme is the dominant one and the subject of this study.

3. Defining the dark and light stripes in terms of wavelength was at first confusing to the reviewer. It is true that the definition of wavelength of the waveform is the wavelength of the lowest nonzero component of the Fourier transform. Calculation is easy for square waves, but there is variation in “duty cycle” as well. It is however described on line 286: “We chose to vary the **wavelength** of the patterns (corresponding to the sum of the width of a blue line and a dark line) while maintaining a constant light-to-dark area ratio of 25% to 75% to keep the global illumination constant.” The reviewer is not sure if there is a better term describing this pattern than “wavelength”.

4. The light intensity measurement is not described (e.g., equipment used, the setup to measure intensity at the correct distance).

Line 25:

Change band pass filter to bandpass filter (use band-pass or bandpass consistently in manuscript).

Line 46:

“... of sucrose occurs extracellularly, the hexoses produced by hydrolysis are public goods as these sugars can also diffuse away and be consumed by adjacent cells (including cheater cells)”

Hydrolysis occurs in the periplasm as mentioned earlier. I would change this sentence to something similar below:

“... of sucrose occurs in the periplasm, the hexoses produced by hydrolysis are not only taken up but also leaked into the extracellular space and become public goods as these sugars can also diffuse away and be consumed by adjacent cells (including cheater cells)”

Line 102:

“Interestingly, the yPH_470 strain only reached ~13% of the invertase activity measured for the yPH_471 strain, despite having the same promoter.”

Is it possible that the YFP fusion interfered with homodimer formation, leading to reduced activity? In addition, it could of course interfere with production and secretion of the enzyme as mentioned in the manuscript.

Fig 3. Legend:

What is the possible explanation of the appearance of a second patch of faster growing cells on the opposite side of the chamber? According to the manuscript those are the “cheaters”. Why is there a non-growing boundary between the two (producers/cheaters)? The reviewer agrees that the spatial segregation at that scale is hard to decipher.

Lines 217 and 218:

Change small e to E? e.g., $\alpha_{\text{heat}} = 1.5\text{E-}25 \text{ mol.s-1.cell-1}$

Line 282:

Change pass-band filter to bandpass filter (use band-pass consistently in manuscript).

Supporting materials:

In general, the figures and tables should be numbered for easy reference. There are missing figure and table numbers in the beginning of the document.

Model of the Cooperator/Cheater system

1. The equations are hard to read in the figure (in general the figure seems to be low resolution).
2. In the table there are inconsistent number representations (e.g., Km1, Km2, Ks, Kme): 0,0008 instead of 0.0008.

Parameter adjustment

1. Please use:

a) $\alpha_{\text{coop}} = 1.8 \cdot 10^{-24} \text{ mol.s-1.cell-1} ::$

$$\alpha_{\text{coop}} = 1.8\text{E-}24 \text{ mol.s-1.cell-1} \quad \text{or} \quad \alpha_{\text{coop}} = 1.8 \times 10^{-24} \text{ mol.s-1.cell-1}$$

b) α_{heat} has the same representation.

Simulation

1. Please correct: 64 Go of RAM -> 64 GB of RAM

Building the OptoCube

1. The https://github.com/Lab513/DIY_OptoCube repo seems to be missing (or still private, which makes sense).
2. The method of measuring light intensity is not described (device, etc.).
3. Supplementary Figure S6 legend: please change $yy_0 = 1,6. 10^4$ and $xx_0 = 1,3. 10^9$ to $1.6\text{E}4$ and $1.3\text{E}9$.

4. Supplementary Figure S7 legend: what is the light intensity (at 100%)? It would be nice to compare it with other studies for phototoxicity.

5. Supplementary Movie SM1. The inclusion of a scale bar in the video is quite tasking. Is it possible to provide the dimensions of the area in the video legend?

Reviewer #4:

Remarks to the Author:

This is an interesting paper on spatial structure and cooperation, which introduces a useful methodology that could allow some elegant manipulations of cooperation/cheating. It uses a well studied yeast system, and develops a strain that allows cooperation to be turned on by light. This is a super cool system, that can allow very precise experiments! The paper is well written - clear and easy to read.

However, while the paper makes a good contribution, I do not think it is suitable for Nature Communications. It is more of a methods (albeit super cool method!) paper than a paper tackling a big novel question for the evolution of cooperation.

1. A major focus of the paper is: "Most studies have focused on competition/cooperation mechanisms in well-mixed populations; thus, it is not known how a spatially structured population". While this may be partially true for yeast (but see Maclean & Gudelj 2006 Nature + related) this is definitely not the case more generally. There is a large theoretical, experimental (e.g. bacteria), observational (e.g. animal) and comparative (animals and microbes) literature on this. Looking at spatial structure, subdivision, relatedness structure, frequency dependence, density dependence, diffusion rates, mating structure etc. Consequently, while this is a good paper, it is overselling its novelty, due to a lack of awareness of the literature on the evolution of cooperation. The paper is not written / developed / framed within the context of the existing literature on the evolution of cooperation. This is especially important given what the paper claims about what has been done before, and makes it hard to judge the novelty of this paper. The most novel aspect of the work is the methodology, not how it is then used.

2. The utility and purpose of the model was not clear, relative to previous theory.

3. Paragraph starting line 341 and related results. This seems to relate to the issue of frequency dependence that has been discussed much in yeast and other microbes?

4. The ending, lines 383-388 feel like an unjustified over-extrapolation.

Referee #1

In this study, the authors use optogenetic control of the invertase enzyme Suc2 in populations of budding yeast distributed in roughly two dimensions to show that glucose diffusion and sucrose diffusion rates determine “cooperator benefit”. Namely, when cooperators and cheaters are distributed at different spatial scales, how much cooperators benefit from their ability to make invertase (Suc2). The authors engineer optogenetic control of expression of Suc2 in budding yeast using the EL222 optogenetic system as well as perform some tuning with an mRNA degradation tag and promoter engineering to identify a variant with the highest dynamic range under light control. There is some nice synthetic biology here, and the authors explain their logic and characterization of the system well. However, it is worth noting (also mentioned below) that Moreno Morales, et al already made a functional, optogenetically controlled Suc2 in budding yeast and the authors, in my opinion, really gloss over this point. The current study’s system may have some benefits over the previously engineered system, but at least make clear that this approach has already been implemented in the literature in the same organism. The authors then use an OptoCube to spatially designate cooperator and cheater populations using patterned light on petri dishes. They vary the size of these populations (this reduces to a 1-dimensional problem due to the thinness of the agar cells are seeded in and the length of the strips such that only width “matters”) and identify a short length-scale cutoff presumably due to glucose diffusion rates and a longer-range cutoff due to sucrose diffusion. In between cooperators receive maximal benefit in the presence of cheaters. This study employs elegant experimental techniques that have not yet been widely applied to study microbial interactions. The choice of invertase expression in budding yeast is a good one, as it is a well-studied system with vast literature. The main finding, of a cooperator benefit bandpass, is interesting but expected in the context of diffusion rates. The study needs to address major comments before it is appropriate for publication.

We thank the referee #1 for his positive assessment of our manuscript. We addressed the different remarks of this referee, and we think that it helped us significantly improve the main message of our article while clarifying its merits compared to previously published works. However, we disagree with the referee’s last statement: to our knowledge, the existence of a cooperator benefit displaying a bandpass behavior has never been reported before and was not “expected in the context of diffusion rates.” We claim it is a novel and important result that we explain through the differential behavior of cooperators and cheaters as a function of their optogenetically controlled spatial assortment.

*The authors state (Line 74) that a method to quantitatively control spatial organization of cooperators and cheaters in a population of microbial cells would enable understanding of how populations of cells utilize shared resources. This is very true. However, the authors gloss over the fact that at least two prior studies have demonstrated this approach. Indeed, that is exactly what is done in Moreno Morales (Ref. 23) using an almost identical approach to the current study, namely optogenetic control of Suc2 expression. While it is true that the current study pushes the spatial study much further and uses a significantly refined technique (namely the OptoCube) for spatial control both spatial control (and patterning) were demonstrated in Moreno Morales, et al. The authors also fail to mention that optogenetic, spatial control of a public good (matrix production) was first demonstrated in the bacterium *Sinorhizobium meliloti* (<https://doi.org/10.1021/acssynbio.0c00498>).*

We thank the reviewer for his/her constructive comment. The reviewer rightfully points out that *Moreno-Morales et al* also used optogenetics using the CRY2-CIB1 light inducible system to activate expression of the invertase SUC2. Yet, as mentioned by the referee, our work proposes an optogenetic strain with better properties (notably, it gives an expression level similar to the wild type's level) and, importantly, we developed a novel experimental device to study quantitatively the spatial interactions of cheating and cooperation. This is a novel (see also response to referee #4) and, in our opinion, elegant demonstration of the potential of optogenetics for the study and the control of microbial interactions. We also thank the referee for pointing out the article by Azady Pirhanov et al. that we were not aware of. We now cite it in our introduction, explaining that while optogenetics is now very popular it has not yet been used extensively (nor to its full potential) in the study and control of microbial cooperation, hence the interest of our work.

Modifications¹

- **Line 94:** “To date, optogenetics has been reported in only a few works as a tool to spatially control public goods, such as the production of extracellular matrix in *Sinorhizobium meliloti*, of adhesin in *E. coli* bacteria or of the SUC2 Invertase in budding yeast. These studies used the potential of spatial patterning of optogenetics to explore the impact on cooperation and diffusion of public goods in microbial systems.
- **Line 99 :** “Here, we build on these ideas, and more specifically on the work by Moreno Morales et al. who showed optogenetic could be used to express the SUC2 invertase in yeast.”
- **Line 112:** The strain construct reported by Moreno Morales et al. was based on the CRY2/CIB1 light inducible system and, although it showed interesting properties, we decided to use the more versatile EL222 light inducible system and performed a set of optimizations to obtain a strain with higher growth rate and induction levels, making it close to the Wild Type behavior growing on sucrose.

The authors do not directly demonstrate that sucrose and glucose diffusion rates are controlling the length scale cutoffs of the cooperator bandpass filter. While the evidence, when combined with the computational model, is certainly suggestive it would be more compelling if they could change diffusion rates using different concentrations of agar, for example. I also feel like there might be a compelling analytical connection here.

Indeed, changing the concentrations of agar would change the diffusion coefficients of glucose, sucrose, and all other nutrients in the gel. The major difficulty with such an experiment is that polysaccharide gels such as agarose and phytigel have large pore sizes with diffusion properties like water. Indeed, Concentrations of 3.8-7.3% agarose reduce diffusion by hindrance. However, effects are evident only for large macromolecules and/or relatively concentrated gels². Therefore, we would indeed have to go to very high and impractical concentrations to hinder a

¹ Line numbers refer to numbers with « all marks » turned on in « track changes » mode

² Johnson et al., Biophys J, 1996. <https://www.ncbi.nlm.nih.gov/pmc/articles/PMC1225002/pdf/biophysj00053-0441.pdf>

small molecule. Indeed, to increase significantly (e.g., by one order of magnitude) the diffusion coefficients by increasing the agar concentration, we would have to experimentally raise the concentration well over 7%. This will lead to faster drying and harder agar gel, which impairs growth (and essentially ruins the experiment). Also, high agar concentrations are likely to increase the physical constraints on embedded colonies and could alter the growth rate and maximum number of cells in each clump of cells. Taking these aspects together, it would be extremely challenging to obtain quantitative information with such experiments.

We can however use the numerical model (even though it is not perfect) to perform a set of simulations with varying diffusion coefficients. This was the goal of Supplementary Figure S10 (now S11) which suggested that the lower the diffusion of glucose the smaller the lower cut-off λ^- and the lower the diffusion coefficient of sucrose the smaller the larger cut-off λ^+ .

Regarding the analytical connection, we can obtain from a dimensional analysis that both cut offs have to scale with $\sqrt{(D_i/k_i)}$, where D_i is the diffusion coefficient of the metabolite of interest and k_i is the rate at which this metabolite is absorbed by cells (cf. main text). Importantly, the scaling is difficult to get because of the dependency of k_i with the sucrose and glucose metabolism. Diffusion is not the only important factor here, which makes sense given that the diffusion coefficient for sucrose and glucose are alike. What our simulations suggest (see above and Supplementary Figure S11) is that the larger cut off scales primarily with $\sqrt{(D_s/k_s)}$ (varying the sucrose diffusion alone has a major effect compared to varying glucose diffusion alone) while the lower cut off scales primarily with $\sqrt{(D_m/k_m)}$ (varying sucrose diffusion alone has little impact on it).

Why do the authors show data using yPH_470 (which has invertase fused to YFP?). It is known from previous work that fusion to invertase disrupts its secretion and in fact, I would image the authors see some growth defects. (Protein accumulation does seem to be apparent in Figure 2C, although it is difficult at this magnification to tell exactly what is happening). As the authors do mention, this is almost certainly the source of difference in function between strain 470 and strain 471.

We thank the reviewer for this comment. Indeed, we agree that showing the yPH_470 strain is not critical for the manuscript (we put it here since we showed all strains made during the optimization process). We have removed results from this strain from the manuscript and just explained that we use a 2A sequence to separate SUC2 from its fluorescent reporter.

Modifications:

- **line 122:** *“It is known that fusion of invertase to a fluorescent protein can mistarget its extracellular localization. We thus used a P2A peptide which allowed us to have a proxy of gene expression levels while keeping SUC2 functional”.*
- we removed the information regarding yPH_470.

The data in Figure 3 seems disconnected from the rest of the paper. While these experiments are visually appealing, I don't think that they reveal any new or quantitative information.

We thank the reviewer for his/her insights. The rationale of this experiment was to first test the capacity of cells to grow within a sucrose environment while monitoring single cell behaviors. This experiment is important to confirm/demonstrate that: at this scale (1) cells do not grow significantly in the dark; (2) we can induce sucrose hydrolysis by yeast and growth locally with light; (3) motion of cells that results from cell growth combined with the long lifetime of SUC2 implies that after a few divisions, cooperators extend far away from the initial area (4) cells in the dark areas can grow by capturing the produced glucose. Overall, this experiment demonstrates that light activation of patterns can work but that even though it is possible to “pattern” areas of “cooperator induced” cells at the 100 μ m scale, this is not a proper setting to infer quantitatively the length scales of cooperation-cheating interactions, due to cell motion. In contrast, With the OptoCube we can test the system at a proper scale (as evidence in Figure 7) and without the limits of cell displacement through growth.

Also, while the caption states that the DMD can illuminate single cells, this functionality does not seem to be used.

The referee is right, although the DMD can target single cells, we did not use this capability in this article, nor will it make sense to use it given the typical lengthscale of the problem (see above). We have now removed this from the caption.

In addition, the authors state that within the device a “few cells” are excited with light to produce invertase. But in fact, looking at the figure it looks like at least 10's of cells, and maybe 100's. (Line 161)

We thank the reviewer for spotting this. We agree that the number of initial cells illuminated in Figure 3 and S4 is of the order of ~400 and ~50, respectively. This was stated line 148 (now 162) of the text. We added the number of cells in the Supplementary Figure S4. For clarity, we removed “small patch” and “few cells” wherever appropriate.

In Figure 4 (C) would it be possible to estimate the actual range of colony diameters? The authors state 10's of microns, but looking at the figure I would say some are on the order of 100's of microns. It is hard to tell from the figure.

From the image set for which an example is shown in Figure 4C, we measured colonies diameter from 10 to 40 μ m. This is now mentioned in the text.

Modification

- Caption Figure 4: “[...] cells develop into microcolonies with diameters ranging from 10 to 40 μ m”

Also, a more interesting question: Do the sizes of these cell clumps affect the spatial dynamics? This is a spatial scale at play in the system, and it is known from Koschwanez, et al to have an effect, but does it affect the authors' results?

We thank the reviewer for this important question. Indeed, the scale of clumps is another scale of the problem. Yet, it is too small to be limiting for diffusion. First evidence comes from the work

of Koschwanez et al. (2011), who used simulations to show that a yeast cell at the centre of a clump exhibits a very similar nutrient intake rate for clumps between 10 and 100 μm . This finding is confirmed by others³ were they found that nutrient penetration is sufficient to support growth for spherical yeast colonies of up to 100 μm of radius. We also previously demonstrated that in 2D colonies, gradient of metabolites can extend to this typical range of a few hundred μm ⁴.

Modification:

- **Line 230:** “embedded yeast colonies grown during our experiments reached size ranging from 10 to 40 μm , small enough so that nutrient diffusion is not limiting within the colony sphere”

Does the seeding matter? IE if you seed less densely presumably the cell blobs can grow larger than if you seed more densely and there is more competition between cooperators?

The referee is right, changing the seeding methods is expected to influence the final cell density and thus the cheater/cooperator interactions. We did not explore this parameter quantitatively, yet we choose a seeding density that led to (1) cell clumps small enough to avoid artifacts (nutrients depleted in the center, mechanical constraints of the gel), and (2) that gave measurable, detectable smooth signals with the scanner of the OptoCube.

Modelling: The model fits between Figure 5A and Figure 5B are quite different. Can the authors elaborate on what they think is missing in the model to generate this systematic error at low light intensities?

The referee is right, the model does not fully capture the experimental results. While the agreement is good for Figure 7 (main experiment of the article), Figure 5 shows several of its limitations. The growth rate variation with light intensity and the maximum density reached at low light levels are not properly accounted for. Currently light intensity is accounted for by a linear dependency of α_{coop} between its max value and its min value (α_{cheat}). To better capture the experimental data, one would need to directly add to the model the influence of light on the expression level of SUC2 which is beyond the scope of this article. Indeed, the model is used to confirm/ guide what is observed in Figure 7 regarding the bandpass filter behaviour of the cooperator benefit, which is done with a constant light intensity and for which the model fits nicely.

It also isn't clear that you see saturation in maximal growth rate in the experiments as is seen in the model? (Perhaps difficult to go higher experimentally).

We agree with the reviewer. This is our mistake, we realized that the Figure 5 displayed in the article was incomplete and missing the 100% data (that were shown in Supplementary Figure S9). We updated Figure 5, improved its readability, and showed that indeed we do not observe a

³ Lavrentovich MO, Koschwanez JH, Nelson DR. Nutrient shielding in clusters of cells. Phys Rev E 2013 <http://dx.doi.org/10.1103/PhysRevE.87.062703>

⁴ Marinkovic et al, eLife, 2019. <https://elifesciences.org/articles/47951>

saturation for the growth rate as we do for the model. This is likely the trace of the limitation of the model to account for the light intensity impact on SUC2 expression, as explained above. Note that this limit has no impact on the rest of our analysis with the model on Figure 7 for which the light intensity is fixed and the agreement between the model and the experiment very good.

In the model, where is there only one cell density term ($dCell$)—doesn't there need to be at least two (cooperators and cheaters) as $dCell$ determines invertase concentration (E)?

The difference between cooperator and cheater cells in the model lies in the invertase production rate (“alpha” parameter), being $1.8E-24 \text{ mol.s}^{-1}.\text{cell}^{-1}$ for cooperator and $1.5E25 \text{ mol.s}^{-1}.\text{cell}^{-1}$ for cheaters (~ ten times less), which in turn determines the invertase concentration. We have now stated this explicitly in the revised manuscript.

Modifications:

- **Supplementary Information, model section:** Cells can be either cooperator or cheater, depending on the presence of light. They have distinct invertase production rate α : $1.8E-24 \text{ mol.s}^{-1}.\text{cell}^{-1}$ for cooperators and $1.5E25 \text{ mol.s}^{-1}.\text{cell}^{-1}$ for cheaters. For intermediate light stimulation α is linearly computed between these two values.

Also is it clear why dE/dt should dependent on hexose concentration? We know that at low glucose concentrations, cells produce a significant amount of invertase.

Indeed, the reviewer is right. The simplest minimal model would be to have no dependency on glucose (no regulation of gene expression with glucose, as we only use 100% light in the relevant experiments shown in Figure 6 and 7). However, such a sensitive (but nevertheless continuous) dependency on glucose concentration allows us to (i) prevent the enzyme to increase indefinitely upon nutrient exhaustion (zero glucose) (ii) have close-to-maximal (constitutive) expression of enzyme at physiological glucose concentrations and (iii) -as the reviewer says - maintain a significant amount of invertase at low concentrations. We therefore think that this use of glucose dependency is well justified.

Modifications:

- **Supplementary Information, model section:** Note that dE/dt (rate of production of the invertase) depends on the hexose concentration following a Michaelis-Menten relationship. Thus, as expected from the literature, a significant amount of invertase is produced at low glucose (50% of the maximal production rate at 0.0002% glucose). Also, this relationship prevents the invertase concentration to increase indefinitely after the nutrients are fully depleted.”

In Figure 6C, why are the areas characterized at $x=10\text{mm}$ and $x=20\text{mm}$. For the cheaters, does it make more sense to characterize their density at a constant distance from the cooperator “edge”?

Following the referee's advice, we measured the final cell density at a constant distance (5 mm) from the edge. This gives a very similar result (see below to the one we showed initially and as

such we decided to keep our initial analysis) which relates better to the analysis done in Figure 7 in which we compare the cell density at the center of both domains.

Caption: Density of cooperators in the center of the illuminated area ($x = 10$ mm) and density of cheaters 5mm away the frontier with cooperators of the profiles in Figure 6B. The final mean density is shown with the solid black line. Concentration values are plotted on a logarithmic scale. Error bars represent \pm one standard deviation for three technical replicates.”

Also looking at the modeling in Figure 6D there seems to be a significant amount of hexose accumulating away from the cooperator domains. Why don't we see more cheater growth?

There is indeed hexose accumulating in the cheater domains, but at low level (~0.004% glucose). We do see cheater cells growing (see the grey line), as expected since glucose is diffusing away from the cooperators. But as it can be seen on Figure 4D, the flux of glucose is reversed between $t=15$ h and $t=25$ h: at first cooperators grow fast and produce glucose that can feed the cheaters, but after some time, the glucose flux is directed back towards the cooperating/cheater frontier where there is the higher cell density. This benefits the cheaters, and the cooperators close to the border but not the cheater and the cooperator far from it. Note we let the simulations run until there is no nutrient left at the end.

Are the models and experiments run to steady state? Or is the timing of image acquisition quite important for determining these effects?

We designed our model with no steady state assumption, but except noted otherwise, all simulations are run until complete nutrient depletion. Experiments are also reaching a steady state relatively early, after ~20 hours and then remain stable over time (e.g. Figure 7C, or Supplementary Movies).

How stable is the cooperator benefit over time?

The cooperator benefit evolution in time can be deduced from Figure 7C by looking at the distance between the cooperator (blue) and cheater curves (grey). The benefit remains stable once the experiments have reached a steady state and we did not observe further growth.

Would it be reduced if cheaters were allowed to grow longer to take advantage of the accumulating hexose?

We are not sure to understand the referee's question. At the end of the simulations/ experiments no residual growth is observed, even in the cheater domains. Indeed, in Figure 7 the cooperator benefit is measured after reaching a steady state.

How the diffusion constants affect the cutoffs is not well described in the main text. There is some additional information and modeling in the supplemental text (S10). But are the bandwidths defined by eyeballing arbitrary cutoffs?

The cut-offs were defined as the distance at which the cooperator benefit is 70% of its maximal value, namely $5.7E8$ CFU/mL which is attained at ~ 5 mm for λ^- and ~ 20 mm for λ^+ . This choice allowed us to define properly both cut offs. The classic definition of the cut offs at "-3dB" (that is 50% of the maximal value) would not allow a precise measure of the upper cut off in our case (it would be too close to the maximum range of the wavelength explored in our setup). As discussed above and reported in Supplementary Figure S11 (previously S10), both cut offs depend on the diffusion coefficients.

Modification:

- **Line 355:** "From the cooperator benefit, we computed two cut-off wavelengths, defined as the distance at which the cooperator benefit is 70% of the maximal cooperator benefit, namely $5.7E8$ CFU.mL⁻¹, which are attained at ~ 5 mm (λ^-) and ~ 20 mm (λ^+)."

Is there a relationship between the diffusion constant and the length scale that is more analytical?

We agree with the reviewer about the interest of finding an analytical relationship for the cut-off wavelength. Both must scale primarily with their respective diffusion coefficient ($\sim \sqrt{D}$), but also with the metabolic properties of the cell (see also above). This second dependency is more complex, and we did not obtain a satisfying (simple) analytical derivation of both cut offs showing the dependency with the absorption of glucose and sucrose. In future work, we will try to further explore such a scaling law numerically.

It is interesting that the averaged population (cooperators + cheaters) is constant across the whole range of light patterns. This conflicts with some work from the Gudelj research group which shows that mixes of cooperators and cheaters are actually more efficient. In fact, reference to the Gudelj lab work, which is extremely relevant to the questions outlined in this study, is conspicuously missing. The authors would do well to situate their current findings with their relevance to this previous body of literature.

We thank the reviewer for mentioning the nice work of the Gudelji's group⁵ (see also our response to referee#4). In MacLean RC et al. (2010), there is indeed evidence that mixes of cooperators and cheaters reaches higher final densities than monoculture on agar plates, with the optimal cooperator frequency being around $\sim 40\%$. Yet, we would like to argue that the

⁵ <https://journals.plos.org/plosbiology/article?id=10.1371/journal.pbio.1000486>

comparison with our work is not that straightforward. Indeed, we are focusing on the (large scale) spatial structure where domains of cheaters and cooperators are spatially separated (not mixed). Indeed, we kept the illuminated area ratio to 25:75; this choice fixes the cheater/cooperator ratio population across experiments, and we do not vary it any further (this would be a nice follow up study). What we do show is that when cheaters and cooperators are studied in well separated domains (which is not at all what is done in Gudelji's article) we observe a band pass filter behavior for the cooperator benefit (that is a heterogeneous growth landscape that depends on the size of the domains). Additionally, we also observed that the average final cell population density remains approximately constant. We do not think it conflicts with others published results for the reason explained above, but also due to the difference of resolution between our method and that of Gudelli's work. Indeed, in Gudelji's work, what is measured is the population fitness which goes from ~ 0.945 to ~ 0.995 , so a $\sim 5\%$ difference: such a difference cannot be measured with our experimental method on cell density (see Figure 7D).

Should the longer length scale depend on cheaters at all? How would this change if there were stripes with no background cheater cells (either with modeling or experimentally—which I realize would be more difficult to do in this system). The discussion casts the upper length scale as having to do with cooperator-cheater interactions, but isn't it really just about the width of the cooperator islands? In general, the no-cheater control is missing from most experiments.

We thank the reviewer for posing these interesting questions. To clarify, we actually did not state that the “longer length scale is having to do with cooperator-cheater interactions”. Instead, we did show that the “higher cut-off wavelength ($\lambda_+ \sim 20$ mm) is due to **cooperator self-competition** for sucrose (see lines 362-363 of the first version of the manuscript)”. Consequently, if there were no cheaters at all between the stripes of “cooperators” (it is unclear if we should still call them cooperators), we should still observe the existence of an upper cut off.

To try to push further the referee's question, we can wonder what would occur for the lower cut off if no cheaters were present. First, it is important to remind that we show in our article that cheaters are indeed cheaters: they are consuming glucose and are reducing the cooperator benefit for patterns smaller than the lower cutoff. If we come back to the referee's suggestion of removing cheaters, we can hypothesize that for short wavelengths, cells and stripe of cells will basically compete for resources (sucrose and glucose). There are no more cooperators, just competitors. As the reviewer points out, performing control experiments without cheater cells between the cooperator domains cannot be done with the experimental method of our article. To try to push this idea, we used numerical simulations (see below).

These new simulations show that, at steady state, the “cheater-less system” also behaves as a band pass filter (black dashed line). The existence of the long wavelength cut-off is not surprising since it is due to cooperator-cooperator competition within cooperator domains (which remains in the absence of cheaters). The lower cut-off also exists but is at least twice shorter than the cut off that is observed in the cheater/cooperator system (cut-off at ~ 2.5 mm instead of ~ 5 mm). It is the trace that when too close to each other cell stripes will compete for sucrose (and glucose). In contrast when cheaters are present between cooperators stripes, the competition for glucose

extends to the “dark domain” and increases the cost for cooperation domains by consuming more glucose (green dashed line).

Taken together, our results confirm that the existence of a lower cut off (~5mm) is the trace of the cheater/cooperator dynamics while the higher cut off is not primarily related to the cheaters as we stated initially in our article. Yet, we do not think that this control is relevant for our study, and we did not add this result in the revised version of our article.

Numerical testing of the impact of the cheater presence on the lower and larger cut-offs of the cooperator/cheater spatial filtering properties.

Please provide more details on the model fitting. The Methods say manual tuning and no fit code is provided in the GitHub code. Was there a loss function used at all? Was additional model validation done?

As stated in the manuscript, we did not use a fitting algorithm. On the contrary, our goal was to demonstrate that this model which is scientifically sound and includes only the main ingredients (and notably diffusion) can be manually adjusted to get a quantitative match with the experimental results of Figure 7. We therefore obtained values for most of the parameters from the literature and manually tuned three of them (as explained in the main text). We clarified this in the supplementary table describing the parameters of the model by indicating the parameters that were “user defined.”

Note that our goal was not to find the best parameters (which could have been done using for example CMAES strategies, even though we would probably have faced overfitting due to the relatively limited experimental datasets). Importantly, the model fits nicely with Figure 7 data

demonstrating that the model captures well the main behavior (and order of magnitude) of the system for high light intensity. Yet, and as it is shown in Figure 5, it should be improved to better represent the maximal growth rate dependency with light intensity and the saturation of cell density at low light intensity (as discussed above and shown in Figure 5). Note that we do not use the model to study the effect of light intensity and only use it as a guide to analyze our experimental results (Figure 7 mostly).

It isn't clear from the supplemental how the issue of light leakage from the DMD with the mirrors "OFF" was solved? It is mentioned as a problem, but what was the solution? And what effect, if any, does this have on your measurements and results if cells are getting some small level of constant illumination?

We mention in the supplement that the DMD light leakage should be minimized (again, the idea is to limit all sources of residual light which might increase the background level of expression of SUC2). This requires using the "sequence mode" of the DMD instead of using a dynamic mode in which images are sent through HDMI from a computer. More details are found in the DMD operating instructions. We thought it was important to mention this, since it took us some time to figure out how to deal with this. The "dark" residual intensity is measured to be 0.0014 mW/cm² which is 1000 times smaller than the 100% setting: it has no relevant effect on the cells, and we (and many other groups) routinely use optogenetics to activate gene expression with the EL222 system without activation at significant level in the dark. Importantly, there is a residual growth in the dark, but not due to a residual illumination (cf. Δ SUC2 slow growth on sucrose). Therefore, residual activity through light leakage is not an issue in our conditions.

Modification

- **Supplementary Information, OptoCube section:** "With this setting, we did not observe significant activation of the EL222 system in the OFF mode."

There is so much theoretical and experimental literature on the invertase regulation system in yeast and its implications for cooperativity in microbial populations. The discussion would benefit from tying the current results to this body of work.

We agree with the referee. It is well known that Invertase production depends on glucose concentration in a non-monotonic manner, with low concentrations inducing expression, and very high concentrations repressing it. However, because the nature of the regulation is transcriptional, and because our strain construction replaces the natural regulation with the PC120 promoter the regulation of the invertase is a level of complexity that escapes the scope of this study. As a follow up study, optogenetic induction of cooperators with native regulation would be an interesting follow up (this could be done using light dependent recombinase while keeping most of the native SUC2 locus).

Minor Comments:

It is not clear from the references or methods what 2A peptide was used. This can affect efficiency greatly (although apparently the efficiency in the authors' case based on function is quite good).

We thank the reviewer for pointing out this missing reference. The 2A peptide used in our work is described as “P2A” in the following article: Liu Z, Chen O, Wall JBJ, Zheng M, Zhou Y, Wang L, Vaseghi HR, Qian L, Liu J. Systematic comparison of 2A peptides for cloning multi-genes in a polycistronic vector 10.1038/s41598-017-02460-2. We used the same name in our article. We added the reference in the text.

A word of caution (Line 287): using wavelength for the width of the illuminated areas might be misconstrued as meaning a different frequency of light.

We thank the reviewer for his/her concern about this possible misunderstanding. However, the term wavelength is appropriate for the band pass filter analogy. For the sake of clarity, we made sure to mention “spatial wavelength” or “wavelength of the pattern” on several occasions in the text.

What is being measured by the scanner is not clear. I am assuming it is just optical density, and that the fluorescence of the cells is not used past the first few Figures. Would be nice to see where the scanner’s capabilities start to saturate as a function of density---what is the range that you are able to measure? And how does this relate to how densely cells grow on the plate? As this is presumably initial seeded densities. Especially since some of the final cell densities seem to be measured at 10^9 (ie Figure 7)—is this within the linear range of the scanner?

The referee is right, the scanner measures reflectance, i.e., the amount of light reflected (by cells in our case). We calibrated this reflectance in Supplementary Figure S6 (now S7). The figure shows how the pixel intensity of a scanned image relates to the cell density of the freshly poured yeast-containing gel. This calibration curve includes high cell density (10^9) and is used to convert the raw image pixel intensity in cell densities in CFU/mL. We clarified this in the method section.

What is low ambient light in the Methods? Is this low light enough to excite EL222 (for some proteins, ie, cryptochromes, it absolutely would be, EL222 may not be as sensitive)

Low ambient light refers to a general avoidance of strong or prolonged exposure to direct light source of the samples. For example, we kept the number and duration of the cell manipulations at the bench as low as possible, working with either a red-light source, or a dim and indirect light source. In our hands, the EL222 system is not activated by such manipulations (see also previous comment on the DMD residual illumination). This is what can be seen from our experiments which all show a clear correlation between illumination and expression of the SUC2 invertase. For clarity we removed the expression “low ambient light” from the methods.

Should you convert Go of RAM into GB of RAM (under “Simulation”, ie French vs English equivalents)? It isn’t that important.

We corrected this.

Can you put a scale bar on Supplementary Figure 8—are these single cells that I am seeing, or clumps of cells? (I’m guessing clumps of cells assuming Figure B is on the same scale as subfigure A?)

The scale bar of the image in Figure S8 (now S9) is indeed the y axis of the plot: Distance in mm. The white dots in Figure S8 are indeed clumps of cells. We have added a scale bar to facilitate the visual estimation of the clump size. We also added the phrase “showing the fluorescence of cell clumps” in the caption of the figure.

The legend of Supp Figure 9 is swapped in terms of which (A or B) is the dark experiment and which is the light experiment.

We corrected this.

Multiple subfigures labels in Supplemental Figure S10 are not consistent with the number of caption descriptions.

We corrected this.

Reviewer #2

*Spatial organisation of microbial communities is a critical aspect in microbial communities' ecology and its understanding can be extremely useful for microbial community engineering and modelling. In this work Le Bec et al., first engineered an optogenetic system that regulates the breakdown of sucrose in the budding yeast *S. cerevisiae* upon light stimulation. They optimise the optogenetic regulation such to diminish the leaky invertase production and be able to fine tune its expression. Then they tested their optogenetic system, first, on a small-scale, using a microfluidic device where they show that spatial organisation induced by their system can rapidly deteriorate upon preservation of *Suc2p* activity in the producer. Then, they developed a system called OptoCube to regulate the optogenetic system activation on a larger scale in petri dish. They demonstrated how carefully tuning of this system can help in understanding and predicting growth of cheater and producer sectors in a controlled setup and showed that both cooperation and competition in the producer sectors can emerge depending on the spatial structure used. I think the work is sound and well executed. I only have few minor comments.*

We thank the referee for his/her positive assessment of our work.

*Line 136, I think that the enzymatic activity of invertase has not been screened in *yPH_536* and *yPH_540* to evaluate its own activity in light and in the dark as has been done with the other strains. Here the authors are referring to maximum difference in growth rate rather than invertase production, which may be expected but not shown.*

We agree these values are of interest, but we did not measure them. We focused our characterization on the **growth properties** of the strains in sucrose, especially for optogenetic strains kept in the dark, which could not be easily deduced from the invertase enzymatic measurement. We have now stated this explicitly in the revised manuscript.

Modifications

- **Lines 161:** “We thus focused our strain characterization on the growth properties in sucrose, as it is the determinant variable dictating if cells are behaving as effective cooperators”

Line 158, should the cells that have been kept in the dark still exhibiting a low growth rate as indicated in Figure 1G?

Cells kept in the dark in a microfluidic chamber may indeed grow, but at a slow rate which is below the threshold of detection of the PIV method at the beginning when there are few cells and little displacement in the image. You can see on Supplementary Movie 1 that cells are growing at a slower pace outside of the illuminated area.

In supplementary Figure S4 the authors refer to the strain yPH_471 in the figure caption but in the main text the paragraph describes the behaviour in the microfluidic device of OptoSuc2 (yPH_536). To which strain is the Supplementary figure referring to?

We thank the reviewer for pointing out this potential confusion. Figure S4 refers to the strain yPH_471, which allows to “see” the optogenetic induction of SUC2 with a fluorescent marker. The main text and Figure 3 refer to yPH_536, the strain we optimized and for which we did not need a fluorescent reporter.

Modification:

- Line 197: “However, cell growth inexorably pushed cooperating cells (i.e., cells that were expressing the Suc2p invertase) away from the illuminated area (Supplementary Figure S3 and also Supplementary Figure S4 showing similar experiment with a YFP SUC2 reporter – yPH_471).”

In Figure 5, 0% and 12% light intensity seem to behave differently in the experimental data compared to the simulated ones. What are the reasons that may explain this different behaviour at low illumination intensity?

We agree with the reviewer. Although the model nicely fit with Figure 7, it fails at reproducing nicely the variation with light intensity, as shown on Figure 5. We think that to capture this dependency, the model should be improved to better consider the dependency of SUC2 expression with light intensity: currently, this is simply done through a linear dependency of the parameter *alpha_coop* with the light intensity. Yet, to build such a model would go beyond the scope of this article since, in addition to the additions of the light dependency in the model, we will need much more data and to use an optimization algorithm (typically CMAES).

Figure 6D, what happens if the simulation is run for the same amount of time of the experiment on plate?

We thank the reviewer for his/her interest in the simulation results for Figure 6. The simulations for single line experiments do yield satisfactory results, although the cooperators self-competition is over-present in the simulation compared to experiments (the Figure 6D corresponds to the yellow line, 5.6 mm.).

Caption: Simulated cell density profiles for single line experiment at $t = 85$ h (no further yeast growth). Cooperator-cooperator-competition is slightly stronger in the simulation compared to experiments.

I could not find the GitHub page for the OptoCube proposed in the supplementary material at the following link https://github.com/Lab513/DIY_OptoCube

Indeed, the Github repository was still in 'private' mode, and it will be open at the time of publication.

Plasmids sequence and primer sequences should be released with the paper.

Plasmid sequences are available as GenBank files on the Zenodo repository that gives access to all data used in this article and that will be open at the time of publication.

Supplementary Figure S9 the figure caption called Figure A and B in the reverse order. A should be the illuminated one and B the growth curves obtained in the dark.

We corrected this.

Line 319 and line 178 please use the same format for referring to the supplementary figure as done elsewhere.

We corrected this.

Legend Figure 3: If I understood correctly the setup of the system in the legend of figure B and C the chambers should be of typically $400 \mu\text{m}$ and not $400 \mu\text{m}^2$ as reported.

We thank the reviewer for pointing out this mistake. We have modified the Figure 3 caption.

In Supplementary Figure S10 the description of panel C, D, E, F is missing.

We have modified the Figure S10 (now S11) caption.

I wonder if there is a specific reason why in Figure 3D the displacement vector map presents a high displacement mainly on the right side of the chamber although the illuminated area encompass a wider part of the device in particular the central part. Is it due to the closeness of the microfluid device side to the illuminated area or is it just a stochastic effect?

There is indeed less high velocity vector at the center of the chamber. This is due to collective cell displacement and the presence of a wall at the bottom of the chamber. Growing cells create a divergent flow which pushes cells outward of the colony centre. And because it is a cumulative displacement, the highest velocities are attained at the edge of the colony. The fact that there is higher velocity on the right than on the left is likely due to stochastic effects / geometric imperfection in the microfluidic chamber. This kind of displacement field is commonly observed for microbial cells growing in a confined microfluidic chamber.

In the simulation paragraph, please change the unit of the RAM used to Gb instead of Go.

We have modified the supplemental.

Reviewer #3

The manuscript describes an interesting method to investigate metabolic interactions between “producer” and “cheater” yeast cells using optogenetic control. What differentiates this study from other similar works studying competition and cooperation mechanisms is that the developed optogenetic quantitative method provides a way to create spatially structured populations with fine control to study behavior and dynamics at the optimal granularity. Combined with an imaging system capturing and tracking cell growth and the developed computational method the authors are able to measure systematically the cost efficiency of the production of a diffusible public good in the cooperating cells and its use by the cheater cells. The authors carefully constructed the yeast system, tested the functionality with a YFP reporter, optimized optogenetic expression of target protein with careful control of expression dynamics (mRNA degradation hairpin). The authors successfully determined optimal band size for producers and cheaters to maximize growth. The study is complete with a model that describes well the observed patterns. The experiments are well designed, and the authors describe the steps taken to design the optimized setup that was used to come up with the final conclusions. The methods are well described with sufficient details for reproducibility, the data available in a public repository (Zenodo), and the diffusion model is accessible on GitHub in a Jupyter Notebook format. This study helps in future endeavors for optimal engineering of designer microbial consortia via optogenetic regulation in biomanufacturing and other areas. In a follow-up study it would be interesting to see collaborator/cheater behavior when circular illumination patterns are used.

The reviewer recommends this manuscript for publication with minor revisions.

We thank the referee for his/her positive assessment of our work.

1. *The authors combine glucose and fructose into hexoses in the modeling and explanation of the experimental outcome. There is a difference between glucose and fructose utilization by yeast under anaerobic conditions. There are many papers describing the preferential use of glucose over fructose, which is especially important in wine making (undesired sweetness of the residual fructose in dry wine).*

<https://doi.org/10.1016/j.femsyr.2004.02.005> The literature is lacking comparative studies under aerobic conditions. Some preference for glucose was identified in aerobic bioreactor runs, but this seems strain specific (<https://doi.org/10.1093/femsyr/foab021>).

We thank the reviewer for pointing out such differences between glucose and fructose yeast metabolism. Our experiments were all performed aerobically and for the sake of simplicity, our model considers glucose and fructose indistinctively, yielding good enough agreement with the experimental results. We now mention this in the model description.

Modifications:

- **Line 246:** While fructose and glucose utilization by yeast can differ in anaerobic conditions, it is unclear what occurs in aerobic conditions and to what extent this would impact the cheating/cooperating dynamics. For the sake of simplicity, we approximated fructose and glucose utilization as identical (i.e. hexose utilization)

It is worth noting that the SUC2 locus encodes two different forms of invertase, the glycosylated homodimer excreted into the periplasm and a cytosolic form. The cytosolic form lacks the signal peptide due to alternative translation. The periplasmic enzyme is the dominant one and the subject of this study.

Indeed, we did not mention the cytosolic form of invertase in our manuscript, we only mentioned in the main text L49 that there is an alternative pathway for sucrose hydrolysis. For clarity, we now explicitly mention the internal form of the invertase.

Modification

- Lines 52: “[...] to then be hydrolysed internally (cytosolic form of invertase, maltase and isomaltase) usually external hydrolysis is the dominant sucrose uptake process in wild-type yeast.”

3. Defining the dark and light stripes in terms of wavelength was at first confusing to the reviewer. It is true that the definition of wavelength of the waveform is the wavelength of the lowest nonzero component of the Fourier transform. Calculation is easy for square waves, but there is variation in “duty cycle” as well. It is however described on line 286: “We chose to vary the wavelength of the patterns (corresponding to the sum of the width of a blue line and a dark line) while maintaining a constant light-to-dark area ratio of 25% to 75% to keep the global illumination constant.” The reviewer is not sure if there is a better term describing this pattern than “wavelength”.

We could have used the term “Spatial period” which is then simply defined as the distance between two identical motifs. Yet, we think that the wavelength wording is better adapted to the analogy with a band pass filter.

4. The light intensity measurement is not described (e.g., equipment used, the setup to measure intensity at the correct distance).

We have modified the manuscript to include the light intensity measurement procedure in the method section.

Modification

- **Lines 550:** "Intensity measurements were conducted using a power meter (TOR Labs PM100D with S120C sensor), placing the sensor at the same position as where the cells would have been growing during the experiment."

Line 25: Change band pass filter to bandpass filter (use band-pass or bandpass consistently in manuscript).

We corrected this throughout the manuscript.

Line 46:

"... of sucrose occurs extracellularly, the hexoses produced by hydrolysis are public goods as these sugars can also diffuse away and be consumed by adjacent cells (including cheater cells)"

Hydrolysis occurs in the periplasm as mentioned earlier. I would change this sentence to something similar below:

"... of sucrose occurs in the periplasm, the hexoses produced by hydrolysis are not only taken up but also leaked into the extracellular space and become public goods as these sugars can also diffuse away and be consumed by adjacent cells (including cheater cells)"

We thank the reviewer for this suggestion. We modified the text accordingly.

Line 102:

"Interestingly, the yPH_470 strain only reached ~13% of the invertase activity measured for the yPH_471 strain, despite having the same promoter."

Is it possible that the YFP fusion interfered with homodimer formation, leading to reduced activity? In addition, it could of course interfere with production and secretion of the enzyme as mentioned in the manuscript.

Yes indeed, we agree with the referee. Note that as requested by referee#1, we have removed mention of the yPH_470 strain since it was not essential to the article message.

Fig 3. Legend:

What is the possible explanation of the appearance of a second patch of faster growing cells on the opposite side of the chamber? According to the manuscript those are the "cheaters". Why is there a nongrowing boundary between the two (producers/cheaters)? The reviewer agrees that the spatial segregation at that scale is hard to decipher.

We indeed observe a patch of faster growing cells at the top of the chamber, i.e., cheaters cells that did not receive optogenetic activation. The upper patch of cells was initially bigger than the neighboring isolated cheaters, so it is the first cheater patch that exhibits significant growth measurable by the image analysis. Indeed, the PIV analysis performs best when the population is sufficiently large, so it can generate significant cell displacements. On supplementary Movie SM1, one can see that some intermediate cells do grow, although slowly.

Lines 217 and 218:

Change small e to E ? e.g., $\alpha_{\text{cheat}} = 1.5E-25 \text{ mol.s}^{-1}\text{.cell}^{-1}$

We corrected this.

Line 282:

Change pass-band filter to bandpass filter (use band-pass consistently in manuscript).

We corrected this.

Supporting materials:

In general, the figures and tables should be numbered for easy reference. There are missing figure and table numbers in the beginning of the document.

We added figures and table numbers in the Supplementary information.

Model of the Cooperator/Cheater system

1. The equations are hard to read in the figure (in general the figure seems to be low resolution).

The figure has been modified to improve the reading of the equations. High resolution figures are available for publication.

2. In the table there are inconsistent number representations (e.g., $Km1$, $Km2$, Ks , Kme): 0,0008 instead of 0.0008.

We corrected this.

Parameter adjustment

1. Please use:

a) $\alpha_{\text{coop}} = 1.8 \cdot 10^{-24} \text{ mol.s}^{-1}\text{.cell}^{-1}$::

$\alpha_{\text{coop}} = 1.8E-24 \text{ mol.s}^{-1}\text{.cell}^{-1}$ or $\alpha_{\text{coop}} = 1.8 \times 10^{-24} \text{ mol.s}^{-1}\text{.cell}^{-1}$

We corrected this.

b) α_{cheat} has the same representation.

We corrected this.

Simulation

1. Please correct: 64 Go of RAM -> 64 GB of RAM

We corrected this.

Building the OptoCube

1. The https://github.com/Lab513/DIY_OptoCube repo seems to be missing (or still private, which makes sense).

We will open the repository as soon as the article is published.

2. *The method of measuring light intensity is not described (device, etc.).*

We added this method (cf. comments above)

3. *Supplementary Figure S6 legend: please change 104 .1,6 = 0 and 109 .1,3 = 0 to 1.6E4 and 1.3E9.*

We corrected this.

4. *Supplementary Figure S7 legend: what is the light intensity (at 100%)? It would be nice to compare it with other studies for phototoxicity.*

We thank the reviewer for pointing out the missing value. 100% intensity corresponds to 1.13 mW.cm⁻². It was stated in the method, but we agree with the referee that it is better to state it again when needed. We have modified the caption.

5. *Supplementary Movie SM1. The inclusion of a scale bar in the video is quite tasking. Is it possible to provide the dimensions of the area in the video legend?*

We added a scale bar in the Movie SM1

Reviewer #4

This is an interesting paper on spatial structure and cooperation, which introduces a useful methodology that could allow some elegant manipulations of cooperation/cheating. It uses a well studied yeast system, and develops a strain that allows cooperation to be turned on by light. This is a super cool system, that can allow very precise experiments! The paper is well written - clear and easy to read. However, while the paper makes a good contribution, I do not think it is suitable for Nature Communications. It is more of a methods (albeit super cool method!) paper than a paper tackling a big novel question for the evolution of cooperation.

We thank the reviewer for the positive assessment of our methodology, choice of model and the manuscript's quality. Although we agree with the reviewer in that we have indeed missed citing important literature—both general and yeast-specific— and did not put our study in this larger context, we think that there is both novelty and—as the reviewer points out—methodological promise for the understanding of spatial microbial cooperation. We consider our manuscript as aiding the field, with a novel, relevant quantitative and precise method for shaping and studying spatial growth patterns and cooperation/ metabolic interactions within microbial systems. On top of that, our study confirms known principles but also yielded unexpected behaviors (see below). We therefore think it deserves consideration by Nature Communications. In general, we have now modified the manuscript to better explain the novelty of our work and we hope our changes will be satisfactory for this referee.

1. A major focus of the paper is: *“Most studies have focused on competition/cooperation mechanisms in well-mixed populations; thus, it is not known how a spatially structured population”*. While this may be partially true for yeast (but see Maclean & Gudelj 2006 Nature + related) this is definitely not the case more generally. There is a large theoretical, experimental (e.g. bacteria), observational (e.g. animal) and comparative (animals and microbes) literature on this. Looking at spatial structure, subdivision, relatedness structure, frequency dependence, density dependence, diffusion rates, mating structure etc. Consequently, while this is a good paper, it is overselling its novelty, due to a lack of awareness of the literature on the evolution of cooperation. The paper is not written / developed / framed within the context of the existing literature on the evolution of cooperation. This is especially important given what the paper claims about what has been done before, and makes it hard to judge the novelty of this paper. The most novel aspect of the work is the methodology, not how it is then used.

We thank the reviewer for this important comment. The quoted phrase indeed might be misleading, and we apologize for not putting the work in the context of the broader literature on the evolution of cooperation. We also thank the reviewer for suggesting the important specific yeast literature we have missed. We have now added a paragraph in the introduction that cites the relevant experimental and theoretical studies he/she refers to and, regarding yeast, the phrase the reviewer cites (in quotes) is rewritten in the revised version such that it refers specifically to yeast, while at the same time citing Maclean & Gudelj (2006) and other related papers.

We agree with the reviewer that, in its current form, the manuscript made it hard to judge its novelty (aside from the novel method, which we agree with the referee has a lot of potential) and

we have now revised the manuscript to make it clear. Specifically, the reviewer asserts that the novel character of our work in terms of exploring spatial structure “may be partially true for yeast”. First, we’d like to emphasize that the work is indeed novel in the yeast system and, second, that, precisely, the choice of the yeast system allows a quantitative description difficult to get in more complex organisms. Importantly, we were able to demonstrate the importance of large-scale spatial structure and how cooperator and cheaters interacts spatially depending on the size of their respective domains of existence. Our analysis of cooperation and cheating dynamics in terms of a bandpass filter allows to summarize these different scales of interactions (including cooperator-cooperator competition and the role of cheaters as a reservoir of sucrose) To our knowledge this has not been described in the yeast invertase system, precisely because methodology was lacking to shape spatially extended domains of cooperators and cheaters. Indeed, a key novel aspect of our work is the capacity to quantitatively vary the lengthscale of cooperator-cheater interactions.

A good example to understand what our approach adds to the field is the (nice) works by McLean and Gudelj (2006 and 2010). Although, these authors studied the structure of a cooperator/cheater population, the structures obtained are very different from the ones we studied. Their spatial structure experiments are obtained by varying the initial frequency of cooperator/cheaters and the population density on cultures that are first mixed and then let to settle and divide. More specifically, in the 2006 work, they addressed the problem of evolution of metapopulations in a 96-well plate format. Indeed, this sort of experiment is interesting to show that isolated populations with specific frequency of cheaters have higher fitness. But there is no analysis, nor control of the spatial positions of cheaters and cooperators within a colony. Thus, it is quite different from what we are doing here: we shape, at large scale, extended domains of “pure” cheater and cooperators, producing cheater-cooperator transition borders and observing their interactions. In contrast, a 96-well plate is typically 5mm in diameter and they are seeded randomly, so in practice (cf. the main results of our article), domains of cooperator benefit, or cooperator self-competition cannot fully appear within a well since we are close or below the lower cut off we identified in our work. **Said differently we propose a novel method to shape spatial assortment at will and** use it to extract the long scale interactions that drive cooperation benefit. Of course, in a follow up project and with our method, we could study the frequency dependence in addition to the spatial structure and check how this compares to the studies mentioned by this referee.

We modified the text to better outline these novel results and make it clear that our goal was not to study the frequency dependence of cooperator/cheater which has already been studied many times, but rather the large scale (mm scale) spatial structure and interactions of the domains in a quantitative way.

Modification:

- Line 55: While it is now well accepted in the literature that spatial structure plays a determinant role in natural communities’ fate; most controlled laboratory experiments pursuing a quantitative understanding on microbial competition/cooperation mechanisms have focused on structures generated by populations with randomly distributed initial cell positions. For example, the seminal works by Maclean et al.

compared the stability of cooperation in yeast cooperator-cheater cocultures, and further used a 96 well plate to mimic the spatial structure of a metapopulations with different frequencies. Although this experimental system was key to study the impact of cheater/cooperator ratio on the global population fitness, it cannot be used to create extended and interacting domains of cheaters and cooperators. In fact, it is hard to experimentally create, in a controlled manner, a microbial community of cheaters and cooperators with a user defined spatial assortment that would allow to explore quantitatively its impact on microbial cooperation. Here, we propose to use optogenetics to solve this issue and further explore experimentally the microbial interactions over scales of spatial assortment.

2. The utility and purpose of the model was not clear, relative to previous theory.

The goal of the model was to guide the interpretation of our experimental results by providing a different method to explore the role of diffusion coefficients and to give us an estimate of the concentrations of sucrose and glucose which cannot be easily measured experimentally with a proper spatial resolution. Note that this is a minimal model, with only limited ingredients and its goal was not to perfectly match all our data (this would require a better implementation of the dependency with light intensity as explained above). Yet, and even though, the model is minimal and most of its parameters are taken from the literature, we were able to obtain a nice agreement with Figure 7 datasets, which is the central result of this article. We also used the model to get a qualitative estimate of the dynamics of the hexose and sucrose concentration profiles (as seen in Figure 6D) and confirmed our interpretations on the diffusion and consumption of the sugars. We have now modified the manuscript to better explain the purpose of the model.

Modifications

- Lines 240: The purpose of this model is to guide the experimental results' interpretation by estimating the spatial variation of sugars concentrations that cannot be easily measured experimentally.

3. Paragraph starting line 341 and related results. This seems to relate to the issue of frequency dependence that has been discussed much in yeast and other microbes?

We thank the reviewer for pointing out the possible confusion with population frequency dependence. Frequency dependence is indeed well known in multiple microbial systems. Typically, cheaters have a frequency dependent fitness: the fitness is high for low cheater frequency (majority of cooperator allow strong cheater growth), and low fitness for high cheater frequency (minority of cooperator is detrimental for cheater growth).

In line 341, we state that the upper length-scale cut-off is due to self-competition of cooperator for sucrose. **Importantly we refer to the experiment Figure 7 which investigates the effect of the size of the cooperator population (spatial assortment) and not their frequency.** In such an experiment the frequency corresponds to the ratio of light area to dark area, here being 25% light 75% dark. As discussed in our answers to referee #1, in our setup, the frequency is fixed. What we vary is the size of the domains of cooperating and cheater cells. Such self-competition

is only possible in spatially structured environment, resulting from the relatively slow diffusion of sucrose which impairs the growth of the centre of large cooperator colonies. This is not due to the frequency-dependence observed in well-mixed liquid cultures. We have now more explicitly explained the difference between population frequency and pattern wavelength.

Modifications

- Lines 336: “We emphasize the fact that the spatial wavelength of the cooperator/cheater pattern does not correspond to the cooperator population frequency, which is here kept constant at 0.25 across all wavelength experiments. Here, we are investigating the impact on the spatial organization of cooperators and cheaters domains, not that of the ratio of cooperator to cheater cells in a well-mixed culture as it is usually done experimentally.”

4. The ending, lines 383-388 feel like an unjustified over-extrapolation.

In the discussion, we humbly extrapolate that our results and method could help in the understanding of the functioning of complex multispecies microbial consortia that are known to be structured in space, such as engineered microbial materials (that are increasingly used in synthetic biology applications), gut or soil microbiota. This only serves as an illustration of the possible benefit of using optogenetics to control spatial structures in micro-organisms consortia within relevant biological applications.

Reviewers' Comments:

Reviewer #1:

Remarks to the Author:

In this revision, the authors have addressed some of my major comments. In particular, there is greater attention to citing the appropriate literature and connecting this work to the vast literature modeling cooperativity and population dynamics using the SUC2 invertase model in *S. cerevisiae*. However, I do think another reviewer's comments regarding connection to the larger literature on spatially structured ecological populations is well taken. The authors have also removed or modified figures/data that were originally confusing, hence clarifying the text.

Other concerns, including the direct demonstration that diffusion rates are controlling the length scales and in general poor agreement of the underlying model have not been addressed. The authors made sound arguments in the rebuttal that this is not critical, nevertheless I think this would greatly strengthen the paper. The paper absolutely extends optogenetic techniques to demonstrate elegant control in patterning microbial communities.

Reviewer #2:

Remarks to the Author:

The authors have satisfactorily responded to the previous comments. However, neither the gitHub nor the plasmid data is publicly available but only stated that will be released upon publication. This prevents the reviewers from checking the details; it is generally a good practice therefore to provide full access at least to the reviewers.

Reviewer #3:

Remarks to the Author:

The reviewer is satisfied with the revision provided by the authors. The utilization dilemma (glucose and fructose) has been explained and suggestions were incorporated into the manuscript.

Reviewer #4:

Remarks to the Author:

The revisions and responses the referees have not changed my opinion on this paper. On the one hand, it develops a very nice method, that the authors should be commended for. On the other hand, it doesn't help advance our understanding of cooperation in nature in a novel and substantial way. The key question is therefore whether the methods advance is sufficient for Nature Communications.

To expand, this is illustrated by the set up / introduction. Some useful text and a couple of references have been added, but it still doesn't read as grounded well within the broader literature. This matters because it makes the experiments feel a bit "we can do this" and very yeast specific rather than a clear outline of a major outstanding problem that needs resolving. Consequently, with regards to our understanding of the evolution of cooperation, the gains felt incremental. Again, I do not want to take away from the technique and methodology – the key issue is their relative importance to the journal.

Reviewer #1:

*In this revision, the authors have addressed some of my major comments. In particular, there is greater attention to citing the appropriate literature and connecting this work to the vast literature modeling cooperativity and population dynamics using the SUC2 invertase model in *S. cerevisiae*. However, I do think another reviewer's comments regarding connection to the larger literature on spatially structured ecological populations is well taken. The authors have also removed or modified figures/data that were originally confusing, hence clarifying the text. Other concerns, including the direct demonstration that diffusion rates are controlling the length scales and in general poor agreement of the underlying model have not been addressed. The authors made sound arguments in the rebuttal that this is not critical, nevertheless I think this would greatly strengthen the paper. The paper absolutely extends optogenetic techniques to demonstrate elegant control in patterning microbial communities.*

We thank the reviewer for this positive assessment of our work and to outline that it reports a useful and novel methodology. We agree with the reviewer that we don't provide direct demonstration that diffusion rates control the cut-off length scales. As we explained in our rebuttal letter, this is not straightforward and are beyond the scope of this article, although we will push these ideas for our future work. For clarity, we have now included a phrase in the discussion stating clearly that no causal relationship has been established:

L358: "Therefore, our results suggest that the diffusion of sucrose (a reserved carbon source for only cooperators) and glucose (a public good for every cell) define the cut-off dimensions of the bandpass filter, however, further experiments are needed to quantitatively test this hypothesis

We also agree with the reviewer that some of the model behaviors are in poor agreement with experimental ones. In the previous revision we gave arguments for the utility of the model, despite these limitations. We apologize for not including these explicitly in the previous manuscript version. We have now explicitly included our argument for the model utility explicitly in the text:

L260: "The gap between experimental data and the model can very likely be improved by improving the model and more specifically by explicitly taking into account the dependence of SUC2 expression with light intensity. However, the model sufficiently replicated the experimental observations (see also Figure 7) for high light intensities, which is the conditions we used for our next experiments.

Reviewer #2:

The authors have satisfactorily responded to the previous comments. However, neither the gitHub nor the plasmid data is publicly available but only stated that will be released upon

publication. This prevents the reviewers from checking the details; it is generally a good practice therefore to provide full access at least to the reviewers.

We thank the reviewer for the positive assessment of our work. The GitHub and plasmid data have now been made publicly available.

<https://zenodo.org/records/7908455> (all datasets, including plasmids maps)

https://github.com/Lab513/DIY_OptoCube

Reviewer #3:

The reviewer is satisfied with the revision provided by the authors. The utilization dilemma (glucose and fructose) has been explained and suggestions were incorporated into the manuscript.

We thank the reviewer for the positive assessment and for helping us improve the manuscript

Reviewer #4:

The revisions and responses the referees have not changed my opinion on this paper. On the one hand, it develops a very nice method, that the authors should be commended for. On the other hand, it doesn't help advance our understanding of cooperation in nature in a novel and substantial way. The key question is therefore whether the methods advance is sufficient for Nature Communications. To expand, this is illustrated by the set up / introduction. Some useful text and a couple of references have been added, but it still doesn't read as grounded well within the broader literature. This matters because it makes the experiments feel a bit "we can do this" and very yeast specific rather than a clear outline of a major outstanding problem that needs resolving. Consequently, with regards to our understanding of the evolution of cooperation, the gains felt incremental. Again, I do not want to take away from the technique and methodology – the key issue is their relative importance to the journal.

We thank the reviewer for stressing the fact that the text is not completely well positioned in the broader evolution of cooperation literature. However, we feel that positioning our work in the context of what is done in microbes is sufficient as, essentially, we are not claiming to answer an outstanding new question in the field of evolution of cooperation, contrary to what the reviewer asserts. We think that the method we developed rather digs into the biophysics of cooperation and, importantly, outlines previously undescribed behavior in the yeast SUC2 system—cooperator-cooperator competition—in quantitative terms. We do think that, even if we only consider the optogenetic patterning method, our work is an important contribution that will help researchers that are interested in how patterns in multicellular assemblies can emerge from metabolic interactions.

Reviewers' Comments:

Reviewer #1:

Remarks to the Author:

The authors have added additional text to clarify that further experiments are needed to prove that diffusion of sucrose and glucose are defining the cut-off dimensions and clarify how the model could be improved. My major concern regarding the work remains understanding what is the major new biological or ecological knowledge gain.

Reviewer #2:

Remarks to the Author:

The github repository is now accessible and thus I have no further comments.

Reviewer #1:

The authors have added additional text to clarify that further experiments are needed to prove that diffusion of sucrose and glucose are defining the cut-off dimensions and clarify how the model could be improved. My major concern regarding the work remains understanding what is the major new biological or ecological knowledge gain.

We thank the reviewer for his constructive comments on our work. We believe that our work will facilitate further studies on the role of spatial organization in the development of microbial ecosystems, providing both an efficient optogenetic method to create spatial assortment of different cell types and a quantitative workflow to study how specific metabolic interactions set the sizes of cell type domains in microbial ecosystems.

Reviewer #2:

The github repository is now accessible and thus I have no further comments.